# Genome-wide fine-mapping identifies pleiotropic and functional variants that predict many traits across global cattle populations

Ruidong Xiang [1,2✉], Iona M. MacLeod[2], Hans D. Daetwyler [2,3], Gerben de Jong[4], Erin O'Connor[5], Chris Schrooten[6], Amanda J. Chamberlain [2] & Michael E. Goddard[1,2]

The difficulty in finding causative mutations has hampered their use in genomic prediction. Here, we present a methodology to fine-map potentially causal variants genome-wide by integrating the functional, evolutionary and pleiotropic information of variants using GWAS, variant clustering and Bayesian mixture models. Our analysis of 17 million sequence variants in 44,000+ Australian dairy cattle for 34 traits suggests, on average, one pleiotropic QTL existing in each 50 kb chromosome-segment. We selected a set of 80k variants representing potentially causal variants within each chromosome segment to develop a bovine XT-50K genotyping array. The custom array contains many pleiotropic variants with biological functions, including splicing QTLs and variants at conserved sites across 100 vertebrate species. This biology-informed custom array outperformed the standard array in predicting genetic value of multiple traits across populations in independent datasets of 90,000+ dairy cattle from the USA, Australia and New Zealand.

---

[1] Faculty of Veterinary and Agricultural Science, The University of Melbourne, Parkville, VIC, Australia. [2] Agriculture Victoria, AgriBio, Centre for AgriBiosciences, Bundoora, VIC, Australia. [3] School of Applied Systems Biology, La Trobe University, Bundoora, VIC, Australia. [4] Cooperation CRV, Arnhem, The Netherlands. [5] CRV Ambreed, Hamilton, New Zealand. [6] CRV BV, Arnhem, The Netherlands. ✉email: ruidong.xiang@unimelb.edu.au

Genome-wide association studies (GWAS) have been widely used in humans[1], animals[2] and plants[3] for at least three purposes: To study the genetic architecture of complex traits; to map and, if possible, identify causal variants for these traits; and to predict the genetic value of individuals for complex traits. Markers from genotyping arrays can map causal variants to a region of the genome, but to identify the causal variants requires whole-genome sequence since the causal variants should be included in the sequence data[4]. Even with whole-genome sequence data, it is difficult to identify the causal variants because their effects are typically small, and there are too many variants in linkage disequilibrium (LD)[5]. These difficulties in identifying causal variants may be reduced by statistical analysis that fits all variants simultaneously[6] and by the use of information on the likely function of genomic sites[7–9]. When these methods are applied to a small genomic region, it is known as fine-scale mapping[10]. However, it would be desirable to fit all sequence variants simultaneously across the genome, i.e., genome-wide fine-mapping. Causal variants often affect more than one trait, i.e. they are pleiotropic, so power to identify them might be gained by multi-trait analysis[11].

Prediction of genetic or breeding value does not need the causal variants to be identified provided variants (e.g. single-nucleotide polymorphism or SNPs) in LD with the causal variants are genotyped and their effects estimated[12,13]. The resulting prediction is known as a polygenic risk score (PRS) in humans and a genomic estimated breeding value (gEBV) in livestock[14–17]. The low effective population size (Ne) in many breeds of livestock causes extensive LD and hence makes this prediction possible with modest numbers of SNPs[18]. However, the predictions are usually far from 100% accurate in humans or livestock. The accuracy depends on a number of factors including the accuracy with which the individual SNP effects are estimated and the extent to which these SNPs explain the genetic variance by their LD with causal variants. The proportion of the genetic variance explained by SNP panels has been estimated to vary from 33 to 90%[19] and this represents a limitation on the accuracy of PRS or gEBV. The accuracy of prediction is typically low when the target population in which the prediction is to be used is different from the population in which the prediction equation was derived, i.e., out-of-sample prediction[20]. For instance, if the populations differ in LD between the causal variants and the genotyped SNPs then the apparent effect of the SNP will not be the same in both populations. Even within a breed LD will change over time due to selection and genetic drift eroding the accuracy of the prediction. The accuracy with which individual SNP effects are estimated is dependent on the size of the training population. Therefore, it would be desirable to maximise the size of the training data, for instance, by combining the training data from multiple populations. Single-step genomic prediction methods[21] that effectively use genotyped and ungenotyped animals can also increase accuracy. However, the gain is offset by the differences in LD between the populations. This problem could be eliminated if the variants that are used in the prediction were the causal variants or SNPs in consistent LD with them across all populations. In this case, the prediction would be more accurate and more robust because the SNPs would explain all of the genetic variance and their effects would be unaffected by changes in LD. In theory we could use whole-genome sequence, or use high-density panels such as HD (~800 K markers)[22] to genotype every animal. However, whole-genome or high-density genotyping of large populations is very expensive. Therefore, a panel with a modest marker number such as 50k, enriched with potentially causal variants that provide a similar genomic prediction power to high-density panels, would be optimal for large-scale genotyping.

The identification of causal variants and accuracy of prediction would be enhanced by a large training dataset representing multiple populations and recorded for multiple traits; (imputed) genome sequence data; a statistical method that fitted all variants simultaneously and which used functional data on these sequence variants. Unfortunately, this is too computationally demanding with current resources. In this paper, we describe a method that approximates this ideal and applies it to data on 44,000 dairy cattle from three breeds (Holstein, Jersey and Australian Red) with records on 34 traits including milk production, fertility, management and body conformation (average $h^2$ of 0.42 ± 0.04 in bulls and 0.16 ± 0.03 in cows). First, we reduce the number of variants from 17 to 1.7 M by carrying out a multi-trait GWAS using single variant regression incorporating the Functional-And-Evolutionary Trait Heritability (FAETH) score, a publicly available ranking of cattle sequence variants based on their functionality and predicted heritability[7]. The set of 1.7 M variants is further reduced to 165k by variant clustering and pruning for LD. Then we carry out an analysis that fits all remaining variants using Bayesian methods. Finally, we derive a set of informative variants that we chose to be designed on a custom 50K array that will enable better genotyping and more accurate prediction of genetic merit in many cattle.

## Results

**Analysis overview.** Our genome-wide fine-mapping analysis utilised two major sources of information: first, the GWAS effects of 17.7 million sequence variants on 34 Cholesky-decorrelated traits[23,24] in bulls and cows (Supplementary Table 1) where a small multi-trait p-value[11] indicates a variant to be pleiotropic and second, the Functional-And-Evolutionary Trait Heritability (FAETH) score[7] where a high score indicates the high functional and evolutionary significance of these 17.7 M variants. Our genome-wide fine-mapping in Holstein (9739 ♂/22,899 ♀), Jersey (2059 ♂/6174 ♀), mixed breed (0 ♂/2850 ♀) and Australian Red breeds (125 ♂/424 ♀) had five major steps as described in Fig. 1.

**Sequence variant prioritisation using pleiotropy and functionality.** The 17.7 million sequence variants were first ranked by their multi-trait p-values divided by their FAETH score[7]. Variants that had low multi-trait p-value and high FAETH score, i.e., pleiotropic and functional, would be top-ranked. Then, those variants ranked within the top 10% of the FAETH adjusted multi-trait p were selected. This led to 1,757,104 variants in bulls with the average multi-trait p being 0.028(±1.78e−05, SE). In cows, 1,756,637 variants were left with the average multi-trait p being 0.022(±1.4e−05). This top 10% of variants in bulls and cows were LD pruned using Plink 1.9[25] to $r^2 < 0.95$ within 5 Mb sliding-windows, leading to 317,804 variants in bulls and 313,760 in cows.

**Variant prioritisation within sliding-window clusters.** If two variants are in LD with the same causal variant, they are likely to show the same pattern of associations with the 34 traits and to be in LD with each other. Therefore, for each pair of variants $i$ and $j$ within a 5 Mb window, we calculated:

$$\rho_{ij} = r_{cor}\left(t_{variant_i}, t_{variant_j}\right) \times r_{LD}\left(variant_i, variant_j\right) \qquad (1)$$

Where $r_{cor}(t_{variant_i}, t_{variant_j})$ was the correlation across 34 traits between the $t$ values (beta/se from GWAS described above) of $variant_i$ and $variant_j$; $r_{LD}(variant_i, variant_j)$ was the LD assessed by the correlation between the genotypes of variant $i$ and variant $j$. $\rho$ was computed for all variant pairs within 5 Mb sliding-windows for variant clustering (see 'Methods'). Within each cluster, the top 50% of SNPs were selected based on their ranking

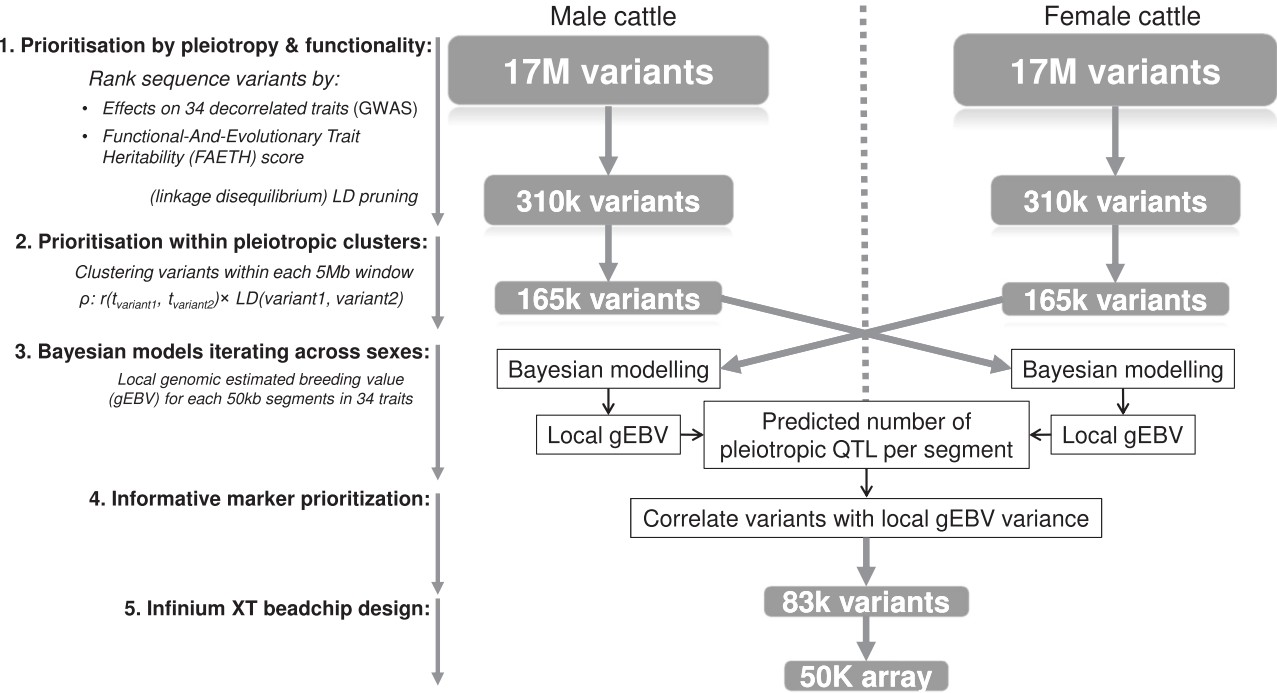

**Fig. 1 Overview of the 5-step genome-wide fine-mapping analysis with 34 traits.** Steps 1–2 that prioritised variants based on the multi-trait GWAS and functional information were conducted separately in the male and female cattle. The design of the Bayesian mixture modelling across two sexes (step 3) aimed at avoiding the bias of variant preselection and is detailed in Supplementary Note 1. Variant effects from Bayesian models were used to calculate local gEBV for each 50 kb segment. Different sets of local gEBV allowed the estimation of the predicted number of pleiotropic QTL per segment. Within each segment, variants with the highest correlation with the local gEBV variance were selected (step 4). These variants were used to customise the XT-50K bovine genotyping array (step 5).

of the FAETH adjusted multi-trait $p$ values. The selection of top variants in each cluster led to 165,214 unique variants remaining in bulls and 164,965 variants in cows. Figure 2a features a small region showing the clusters on chromosome 5 and genome-wide results are provided in Supplementary Fig. 10.

If each cluster represents a single, independent QTL then when the most significant variant from a cluster is fitted in the statistical model, the significance of other variants in that cluster should drop while variants in other clusters should not. We tested this by repeating the multi-trait GWAS but jointly fitting the top 100 variants (tagging up to 90 clusters) from the 165k variant prioritisation. The results in a region of chromosome 5 (Fig. 2b) and across the genome (Fig. 2c) showed that only the variants in the same clusters as the 100 jointly fitted variants dramatically dropped in significance. These results support the suggestion that each cluster represents a pleiotropic QTL.

**Bayesian mixture modelling across sexes.** The next step used BayesRC[6] to model the effects of all 165k selected variants simultaneously with their functional priors, where the variants were divided into three categories defined by their FAETH score 'high' (top 1/3 of the FAETH score), 'medium' (middle 1/3) and 'low' (bottom 1/3) (see 'Methods'). If BayesRC is applied to the same data as used to select the top 165k variants, this can result in bias because the non-significant variants are missing from the BayesRC analysis. To avoid this bias of variant preselection, we analysed the 165k variant set discovered in the bull analysis in the cows and vice versa (Fig. 1 and Supplementary Note 1). The BayesRC analysis resulted in a prediction equation that predicted phenotype (breeding value) for each trait from the 165k variant genotypes. This prediction equation was also applied to the variants within each 50 kb chromosomal segment to generate the genomic estimated breeding value for each 50 kb segment, i.e.,

local gEBV, for 34 traits. Chromosome segments had on average 4.0 variants in bulls (ranging from 1 to 67) and 4.2 variants in cows (ranged from 1 to 83). In total, up to 16,524,586,466 records of bull local gEBV (40,763 segments × 11,923 bulls × 34 traits) and 43,577,017,830 records of cow local gEBV (39,635 segments × 32,337 cows × 34 traits) were generated for further analysis.

If a chromosome segment contained causal mutations for a quantitative trait, the predicted breeding value using this segment would display a large between-individual variance. The variance of local gEBV across individuals has previously been shown to be a useful metric for prioritising informative genomic regions[26] and was used here. Across 34 traits, the average local gEBV variance was $1.6e-06$ ($\pm 3.4e-08$) in bulls with the maximum variance being 0.062. In cows the average local gEBV variance was $1.3e-06(\pm 2.9e-08)$ with the maximum variance being 0.048. Note that the trait variance was close to 1 after Cholesky transformation. The local gEBV variance for each trait is summarised in Supplementary Table 2.

The bias that occurs when the same animals are used for selection of the most significant variants and for the BayesRC analysis occurs in the estimated mixing proportions of the BayesR model. Therefore, an unbiased BayesR analysis in the bulls using the variants selected in the bulls can be obtained if the mixing proportions needed for the BayesR model are estimated in the cows. This approach is used in the next section.

**The number of pleiotropic QTL per segment by analysis of (co) variance of local gEBV.** The BayesRC analyses were carried out one decorrelated trait at a time so if a chromosome segment has a large variance of local gEBV for multiple traits, this could be due to a single pleiotropic QTL or to multiple QTL each affecting a different trait. If it is due to a single pleiotropic QTL then the

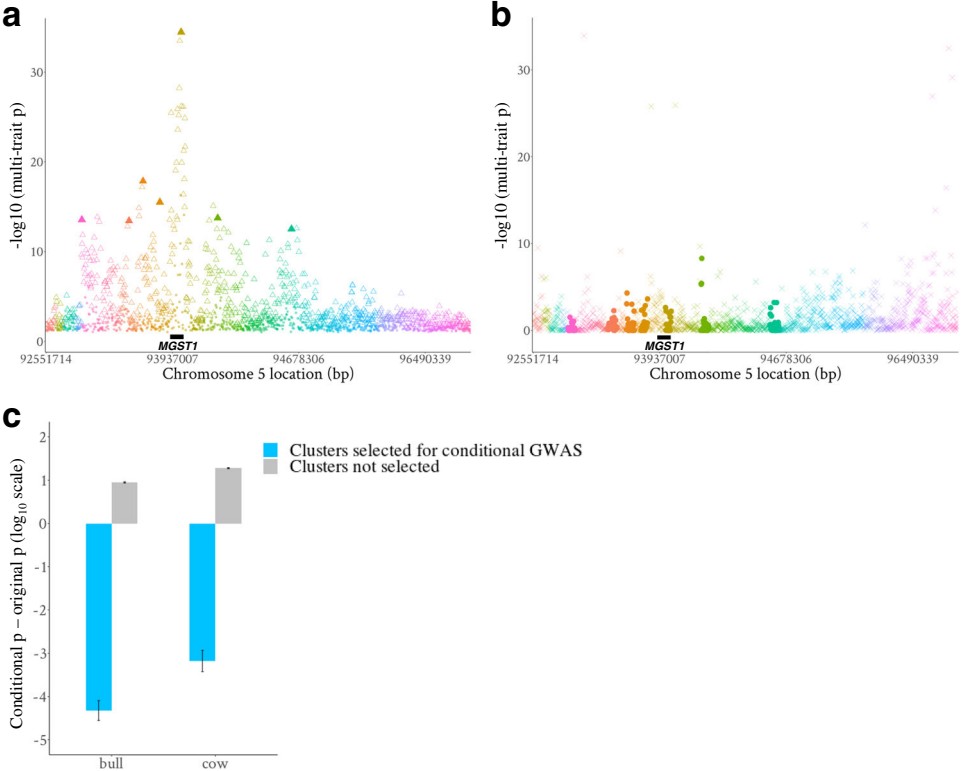

**Fig. 2 Examples of variant clustering and prioritization and results of the conditional analysis. a** Manhattan plot of the sliding-window containing the *MGST1* gene. The *y*-axis is the multi-trait *p*-value of 34 bull traits in this region. Different symbol colours indicate variants in different clusters. Empty triangles are those variants within the top 50% ranking of multi-trait *p*-value adjusted (divided) by FAETH score. These variants across all sliding-windows made up the 165k prioritized variants in both sexes. The solid triangles are among the top 100 variants that appeared in both bull and cow set of prioritized 165k variants and these 100 variants were fitted as covariates in the GWAS for conditional analysis. **b** Manhattan plot of the conditional analysis of the window containing the *MGST1* gene. The *y*-axis is the multi-trait *p*-value of the conditional GWAS of 34 bull traits when fitting the top 100 variants in the model (i.e., the solid triangles in panel **a**). Different symbol colours indicate membership of variants to clusters. Solid circles are the variants from those clusters in which the top 100 variants (e.g., solid triangles in **a**) were selected, while crosses represent variants from those clusters that did not include a top 100 variant. **c** A bar plot showing the genome-wide difference of *p*-values between the conditional GWAS and of the original GWAS after combining sexes. The standard error bars of the mean are across more than 30,000 clusters selected and not selected for conditional GWAS.

animals should rank in the same order on local gEBV for all traits; whereas if there were different QTL for different traits in this segment, then the ranking for each trait should be different. To test whether one or more QTL occurs per chromosome segment, we performed an analysis of two sets of local gEBV. One set was estimated by the aforementioned BayesRC where the variant-predictors were trained in the opposite sexes (e.g., markers pre-selected in bulls and trained in cows called the 'cow equation'). The other set was estimated by BayesR where the variant-predictors were trained in the sex with which the variant-predictors were prioritised (e.g., markers pre-selected in bulls trained in bulls called the 'bull equation'), but using the Dirichlet prior ($\alpha$) for the distribution of variant effects[19] from the BayesRC runs in the opposite sex (see Supplementary Note 1 and Fig. 3a). We calculated a weighted correlation (Fig. 3b):

$$r_{\text{weighted}} = \frac{\sum_i^n \sum_{j(i \neq j)}^n |C_{ij}|}{\sum_i^n \sum_{j(i \neq j)}^n \sqrt{V_{ii} \times V_{jj}}}. \quad (2)$$

$\sum_i^n \sum_{j(i \neq j)}^n |C_{ij}|$ was the sum of the absolute value ('Methods' and Supplementary Note 2) of the off-diagonal elements of the matrix ($C_{ij}$ corresponded to squares labelled by grey 'C', i.e., covariance, in Fig. 3a). $\sum_i^n \sum_{j(i \neq j)}^n \sqrt{V_{ii} \times V_{jj}}$ was the sum of the pairwise geometric mean of the diagonal elements (where $i \neq j$, $V_{ii}$ and $V_{jj}$ corresponded to squares labelled by red 'V', i.e., variance, in Fig. 3a). The number of QTL per segment was estimated as

$1/r_{\text{weighted}}$. As a result, the average number of QTL per segment was estimated to be around 1 in both sexes ($0.91 \pm 0.001$ in bulls and $0.92 \pm 0.001$ in cows, Fig. 3c, d). In fact, most 50 kb chromosome segments contained a single QTL or at least one QTL that dominated the local gEBVs for all traits. Therefore, the strategy of selecting variants that tagged one QTL per segment was adopted as described in the following.

**Selection of 80k variants that best explain local gEBV variance.** We wish to identify a reduced panel of variants that could be used to predict the breeding value of all 34 traits. To do this we looked for variants within each chromosomal segment that were highly correlated with the local gEBV for all traits by calculating:

$$\text{Var}_{g_{\text{local}}}(\text{variant}) = \text{Var}(g_{\text{local}}) \times r^2(\mathbf{g}_{\text{local}}, \mathbf{x}) \quad (3)$$

where $\text{Var}_{g_{\text{local}}}(\text{variant})$ was the amount of local gEBV variance explained by each variant, $\text{Var}(g_{\text{local}})$ was the local gEBV variance, $r^2(\mathbf{g}_{\text{local}}, \mathbf{x})$ was the squared correlation between the vector of local gEBV ($\mathbf{g}_{\text{local}}$) and the vector of the genotype allele count of the variant ($\mathbf{x}$). The sum across 34 traits, $\sum_1^{34} \text{Var}_{g_{\text{local}}}(\text{variant})$, was calculated for each variant and used to select variants ranked in the top 3 in both sexes per segment. This led to the selection of 80k variants (83,455) as shown in Fig. 1. The original multi-trait GWAS *p*-value of these 80k variants are shown in Supplementary Fig. 11.

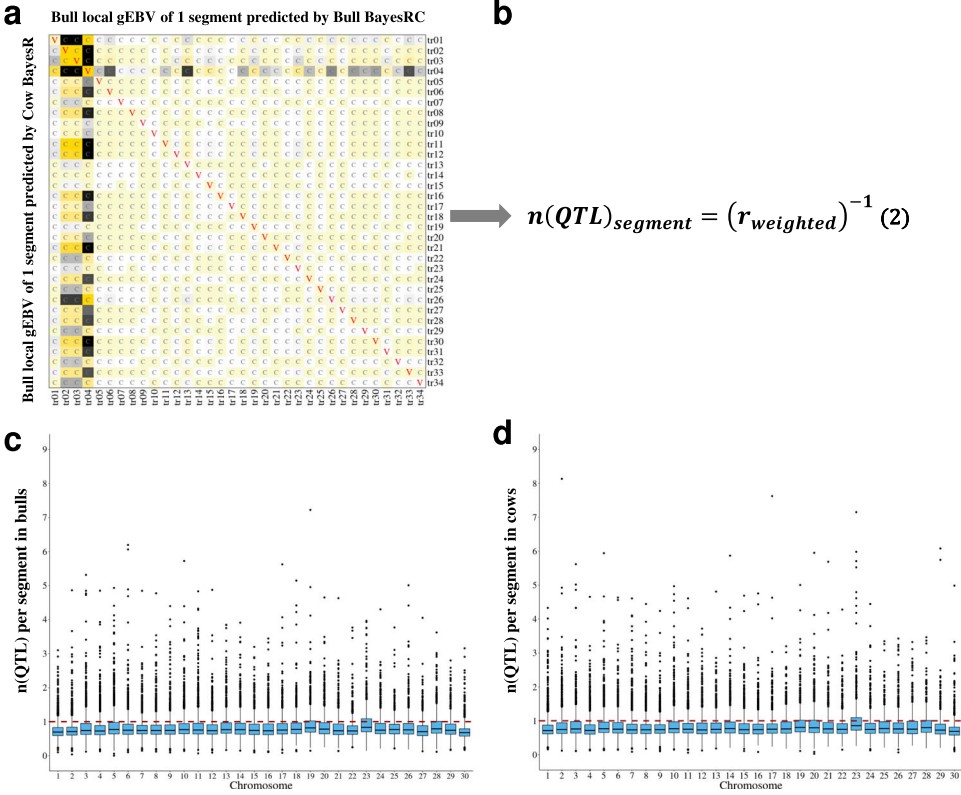

**Fig. 3 The estimation of the indictive number of pleiotropic QTL across 34 traits (tr01–tr34) for each chromosome segments. a** An asymmetric matrix of the variance and covariance of local gEBV of 34 traits for a chromosome segment. The asymmetry (the colour difference between elements above and below the diagonal) of the matrix is due to the use of two sets of local gEBV to calculate the covariance from different training populations (bulls or cows). The yellow colour indicates positive values and the dark colour indicates negative values. The diagonal elements are labelled as red 'V' (variance) and the off-diagonal elements are labelled as grey 'C' (covariance). **b** Equation (2) uses the inverse of the weighted correlation ($r_{weighted}$) of the asymmetric variance and covariance matrix (panel **a**) to estimate the number of QTL within each segment ($n(QTL)_{segment}$). **c** boxplot for the estimation of the number of QTL across 39,635 chromosome segments in bull data. **d** Boxplot for the estimation of the number of pleiotropic QTL across 40,763 chromosome segments in cow data. Each dot on **c** and **d** indicate the estimated number of causal QTL per segment. For each box, the minimum is the lowest point, the maximum is the highest point, whiskers are maxima 1.5 times of interquartile range, the bottom bound, middle line and top bound of the box are the 25th percentile, median and the 75th percentile, respectively.

**Illustration of mapping and fine-mapping of QTL.** The results of GWAS, posterior probability (PP) of BayesRC and the metric of $Var_{g_{local}}$ (variant) were compared in the region of genes *MYH9-CSF2RB* (Fig. 4). Previously it has been suggested that this region harbours multiple QTL[6,27]. In Fig. 4a, where the results of single variant GWAS are plotted, it is not clear how many QTL occur in this region and no variants were genome-wide significant. Figure 4b shows the posterior probability (PP) of individual variants having a non-zero effect. No variant was included in the model 100% of the time but some were included in over half the iterations. Figure 4c accumulated the PP of variants over a 50 kb segment. There were multiple segments in which at least one variant was included in the model almost every iteration. This implied multiple QTL in this region which is supported by the multiple clusters defined by the dot colouring in Fig. 4a. That is, there appear to be several QTL in this region, but the data does not clearly indicate which variants were causal. We wish to select a reduced number of variants to design a custom array for future genotyping and Fig. 4d shows the basis for this selection. That is, the variants were selected based on their correlation with the local gEBV in each 50 kb segment. Subsequently, by ranking variants using multi-trait $Var_{g_{local}}$ (variant) in each segment (right panel), the final set of 80k variants was prioritised. As shown in Fig. 4e, many prioritised 80k variants were also functionally important, including several variants as splicing QTL (sQTL)[28] affecting

excision ratio of introns in *APOL3*, *NCF4* and *CSF2RB*. Three prioritised variants in this region also had $p < 0.05$ for a 36-trait GWAS in US Holstein cattle[29]. A more systematic analysis of the enrichment of US GWAS signals in the Australia-prioritised variants is presented in later sections.

A further illustration (Supplementary Note 3) of the prediction accuracy of 8 types of variant selection using data of 6 traits from 42.2k cows across 3 breeds supported our method of variant selection using local gEBV from BayesRC. In addition, across all scenarios, the FAETH ranking[7] based variant selections showed competitive performances in prediction accuracies, while GWAS based top variant selections had the worst prediction accuracies (Supplementary Note 3).

**Final selection for inclusion on custom Infinium XT-50K beadchip.** The prioritized 80k variants were used as a base for the array design as developed using the systematic approach described (Supplementary Fig. 12). Variants (SNPs and INDELs) were submitted to DesignStudio (Illumina Inc.) according to manufacturer instructions and variants with a design score >0.4 were selected. Attention was given to ensure relatively even distribution across the genome, a preference for Infinium II beadtype (where possible) and variants that were both prioritised by us and appeared in previous standard SNP chips[30]. As a result, over 46.5k sequence variants were selected as the final design for the

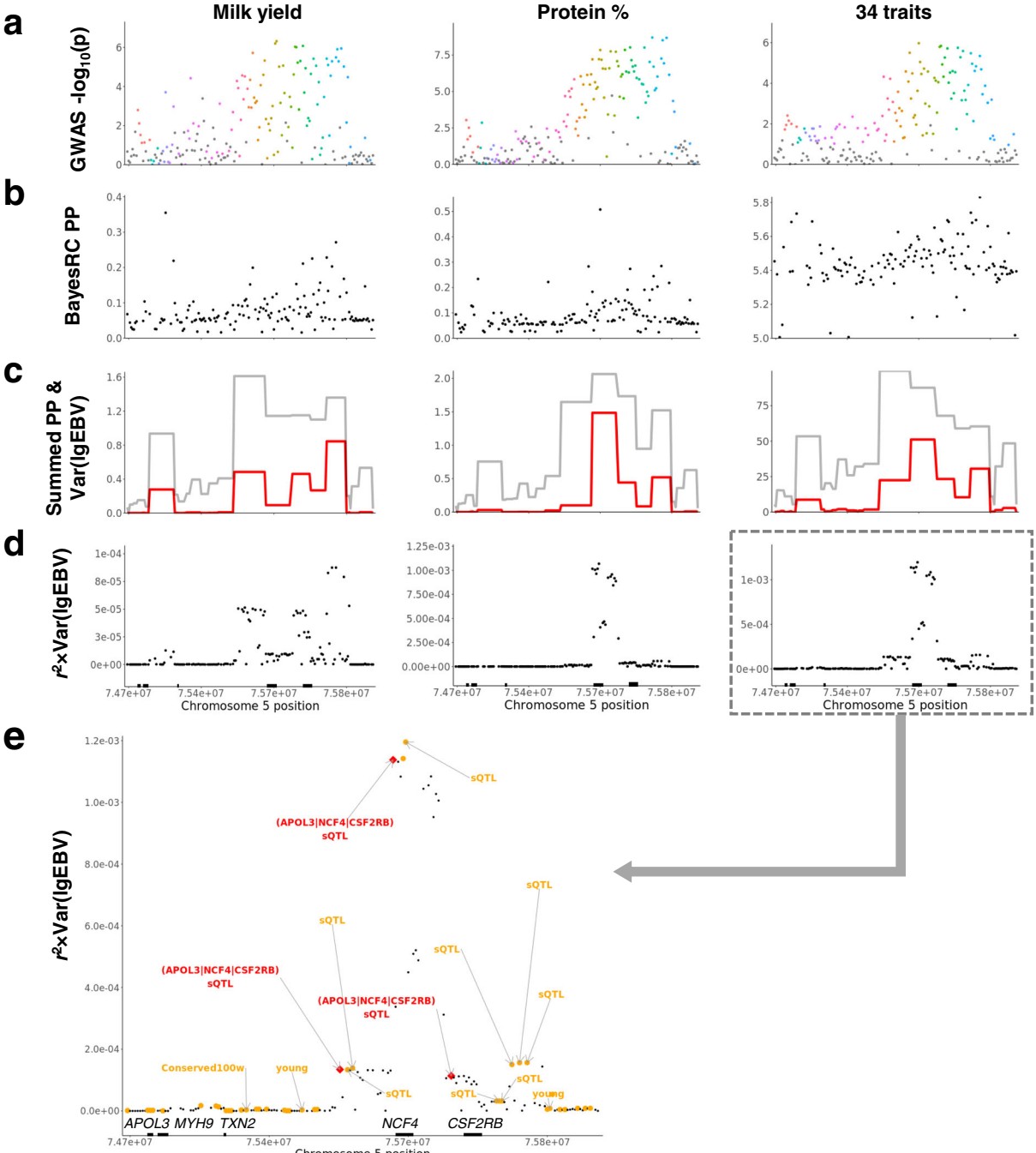

**Fig. 4 Illustration of fine-mapping results in the region of *MYH9-CSF2RB*. a** Manhattan plot of the single-trait GWAS of milk yield (left) and protein percentage (middle), and multi-trait meta-analysis GWAS of 34 traits (right). Dot colour indicates variant membership to different clusters (putatively representing independent QTLs) defined by Eq. (1). **b** Posterior probability (PP) of individual variants from BayesRC runs of milk yield (left) and protein percentage (middle) and the sum of PP across BayesRC runs of 34 traits (right). **c** Summed PP of variants within each chromosome segment (grey line) and the variance of local gEBV for each segment (red line) for milk yield (left), protein percentage (middle) and for 34 traits combined (right).
**d** $\mathrm{Var}_{g_{local}}$ (variant) for milk yield (left), protein percentage (middle) and for 34 traits combined (right). Black bars on the x-axis are genes labelled in (**e**). **e** FAETH[7] annotated data of the right panel of (**d**). Large orange points are variants prioritized in the 80k selection and large red diamonds are variants prioritized in the 80k as well as having $p < 0.05$ in the US GWAS. Three variants that are prioritized by the Australian 80k selection and have $p < 0.05$ in the US GWAS are also splicing sQTLs in an Australian RNA sequencing dataset[28]; the target genes (shown in red colour parenthesis) of these sQTLs have altered intron excision ratio (FDR < 0.05). Variants in the region that are conserved across 100 vertebrate species (conserved100w) and are historically young[7] are also labelled.

custom 50 K array (termed as 'XT-50K' array in the following text). The average gap between markers was 57.1 ± 0.4 kb, which was comparable to 65 ± 0.4 kb for the Standard-50K panel. As shown in Supplementary Fig. 13, on average, the XT-50K variants

were as common as the standard-50K SNPs in the Holstein breed, and were a slightly rarer than the standard-50K SNPs in Jersey and Australian Red. However, the XT-50K variants were much more common than random sequence variants in all breeds.

**Table 1 Summary of the markers on the XT-50K genotyping array.**

| Variant type | Count |
|---|---|
| SNP | 45,226 |
| INDEL | 1285 |
| Gene body | 18,569 |
| Regulatory | 13,627 |
| Evolutionary | 2157 |
| Multi-trait $p < 0.05$ in either sex | 39,756 |
| Multi-trait $\pi > 0.9$ in either sex | 37,503 |
| Low MAF | 3373 |

Gene body: not annotated as 'intergenic' from Ensembl VEP[31] and NGS-variant[32]. Regulatory: milk fat mQTLs, sQTLs, aseQTLs, eeQTLs, geQTLs and variants under ChIP-Seq peaks. Evolutionary: variants at sites conserved across 100 vertebrate species and/or under selection. Multi-trait $p$: $p$-value of the multi-trait meta-analysis (chi-square test[11]) of 34 single-trait GWAS in each sex. Multi-trait $\pi$: multi-trait posterior probability of non-zero effects across 34 traits as summed across single-trait BayesRC runs. Low MAF: variants with minor allele frequency <5% in both sexes.

The final selection included 45,226 SNPs and 1285 INDELs (Table 1). Around 40% of the XT-50K markers (18,569) were non-intergenic variants[31,32] (gene body variants), including categories (originally from Ensembl and NGS-SNP[31,32]) of 'coding.related', 'noncoding.related', 'splice.sites', 'UTR', 'gene. end', and 'intron'[7]. Around 30% (13,627) were putative regulatory variants from our FAETH analysis[7,28,33], including milk fat metabolite QTLs (mQTLs), splicing QTLs (sQTLs), allele-specific expression QTLs (aseQTLs), exon expression QTLs (eeQTLs), gene expression QTLs (geQTLs) and variants under ChIP-Seq peaks in multiple tissues. Around 5% (2158) were involved in within and/or across species evolutionary processes, including 578 variants under selection and/or young and 1604 variants within sites conserved across 100 vertebrate species. Over 80% of the markers (39,756) had an effect on more than 1 out of 34 traits in at least one sex (multi-trait $p < 0.05$, Table 1). Over 3.3k markers had relatively low minor allele frequency (MAF < 5% in both sexes) in the Australian population. All the low MAF markers which were newly discovered (not pre-existing) had a multi-trait $p < 0.05$ for GWAS of 34 traits in at least one sex and 99% of them had a multi-trait PP ($\pi$, single-trait PP summed over 34 traits, see 'Methods') of non-zero effects across 34 traits >0.9 for BayesRC mapping.

**Validation of the XT-50K custom array for genomic prediction in global cattle data.** Three validation tests were conducted. The first test used an independent dataset of 28.2k multi-breed Australian cows to train the BayesR prediction equations for three milk production traits using: the custom XT-50K array, the standard-50K array and the combined markers from the XT-50K and standard-50K arrays. These Australian cow genomic predictors were used to predict the milk production phenotype of 21.2k New Zealand purebred and crossbred cows (Holstein and Jersey breeds). Note that cow phenotypes usually have high error variances and low heritability[7,26]. The XT-50K markers increased genomic prediction accuracies, defined as the correlation between the gEBV and cow phenotype (same below), in the New Zealand cows compared to the standard-50K markers in all scenarios (Fig. 5a). Averaged across 3 traits, the relative increase of the prediction accuracy for the XT-50K from the standard-50K ($\frac{[r_{XT-50K} - r_{Standard-50K}] \times 100\%}{r_{Standard-50K}}$, same below) was 9% for pure Holstein, 11% for pure Jersey and 7.9% Holstein–Jersey crossbreds. In all scenarios, the performance of XT-50K markers was not significantly different from the markers combined from two arrays (Fig. 5a).

The second validation test used the estimated variant effects from a 36-trait GWAS on over 27.1k Holstein bulls from the United States[29]. The US study used 2.7 M imputed sequence variants from Run 5 of the 1000 Bull Genomes Project[4,34] and excluded intergenic and intronic variants[29]. Thus, from the US study variants, there were 27.7k variants overlapping markers on the XT-50K and these were used to predict 3 milk production traits of 28.2k Australian cows and of 21.2k New Zealand cows (Fig. 5b). In most genomic prediction scenarios (17/18), the XT-50K markers increased genomic prediction accuracies compared to the standard-50K markers. Averaged across the 3 traits in the Australian population, the relative increase of the prediction accuracy for the XT-50K from the standard-50K was 1.4% for Holstein, 60% for Jersey and 30% for the Australian Reds. Averaged across the 3 traits in the New Zealand population, the relative increase for the XT-50K was 14% for pure Holstein, 90% for pure Jersey and 16% for Holstein–Jersey mixed breeds. Genomic prediction accuracies of Jersey fat with the standard-50K markers were very low.

The third validation of the XT-50K markers and standard-50K markers was to test for their enrichment in pleiotropic variants in the US GWAS study (meta-analysis using method of ref. [11]) associated with up to 36 USA traits[29]. Only 19 of 36 US traits were found in the 34 Australian traits that were used to prioritise the XT-50K markers (Supplementary Table 3). At any $p$-value threshold of the multi-trait meta-analysis of GWAS, a small $p$-value of which indicates a variant to be associated with many traits[11], variants affecting at least one of the 36 US traits were strongly enriched in the Australian XT-50K markers and such enrichment of pleiotropic variants was always much higher than that of the standard-50K markers (Fig. 5c). The US multi-trait $p$-value in comparison with the Australian multi-trait $p$-value for the XT-50K markers is published (see 'Data availability').

Further, the predictive power of markers from our XT-50K panel was compared to existing panels including the standard-50K (50k SNPs), GGP-F250 (up to 250k SNPs)[22] and high-density (HD, up to 600k SNPs)[26] using data of 6 traits recorded in 42.2k Australian cows across 3 breeds (Supplementary Note 4). We showed that the predictive power of the XT-50K, with a much smaller number of markers, was similar and often better than denser panels such as GGP-F250 and HD. The standard 50 K panel had the lowest prediction accuracy across all scenarios.

## Discussion

In this paper, we have demonstrated a method that fine-maps genome-wide informative sequence variants with pleiotropic effects and functional significance. Tested using global datasets, we show that a selection of <50k markers from these informative variants can be used to predict multiple traits in populations quite different from the one used for training the prediction equations. Genomic prediction tested in this study using our XT-50K panel which contains up to 50k potentially causal markers, is as powerful as dense panels such as GGP-F250[22] and HD that contain hundreds of thousands of markers. This implies that by using the XT-50K panel in future routine genotyping, good prediction accuracy can be achieved with a lower cost genotyping platform that is affordable to farmers. Our results based on real cattle data contrast the results from Karaman et al.[18] using simulated human data which suggested that the selection of informative variants has little benefit in increasing genomic prediction accuracy. Significant differences in the types of genomic and phenotype data, models, and design used between the two studies could lead to different conclusions. However, our conclusion is supported by extensive discovery and validation analyses using data of over 100k cattle with multiple traits across breeds and countries. The

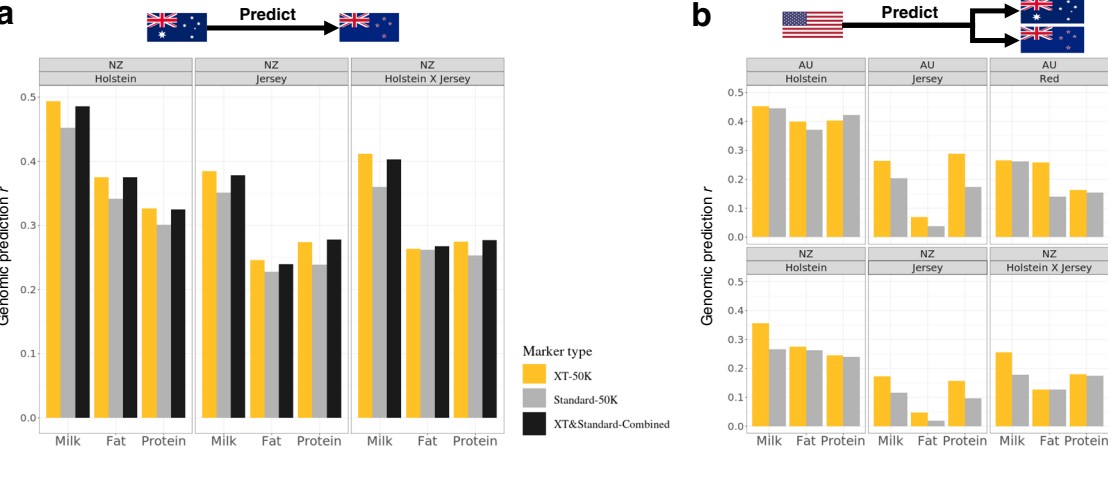

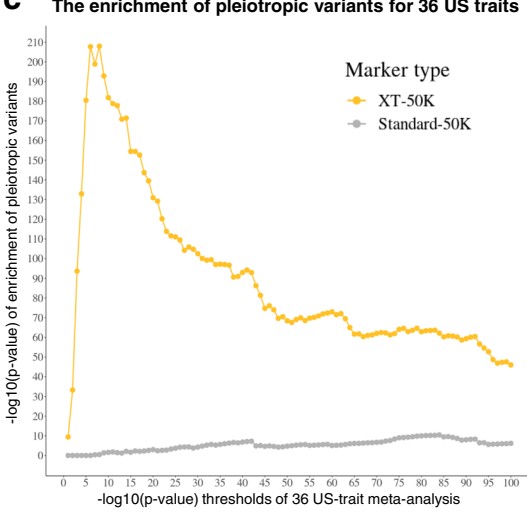

**Fig. 5 Comparing customized XT-50K markers with the Standard-50K markers in multiple validation cow sets. a** BayesR genomic prediction trained in an independent set of 28.2k multi-breed Australian cows for three traits and predicted into 21.2k New Zealand cows (pure- and crossbreds). XT&Standard-Combined: combined markers from the XT-50K and Standard-50K panels. **b** Genomic prediction using a subset of markers from the two 50 K panels that are also found in the US 36-trait GWAS using 27.1k Holstein US bulls for the same three traits and predicted into 28.2k Australian cows and 21.2k New Zealand cows. The US GWAS excluded intergenic and intronic variants[29] and their sequence variant imputation was based on Run 5 of the 1000 Bull Genomes project[4]. **c** The enrichment of pleiotropic variants from the US 36-trait GWAS in the two 50K panels. Note that only 19 of 36 US traits were present in the 34 Australian traits used to prioritise the XT-50K markers (Supplementary Table 3). X-axis: multi-trait p-value thresholds (0–100 on −log$_{10}$ scale) used to select significant pleiotropic variants for USA traits. Y-axis: the significance of enrichment (hypergeometric test) of pleiotropic variants for USA traits based on the number significant variants at each multi-trait p-value threshold (X-axis) within the XT-50K markers and within the Standard-50K markers.

use of the dairy cattle data in our study takes advantage of large datasets from this sector, like previous studies[6,7,19]. The methods presented in the current study can also be applied to other species.

The main feature of our variant prioritisation is the incorporation of functional information of variants not necessarily correlated with their size of effects on complex traits. Besides, variants were selected if they were associated with at least one trait (uncorrelated multi-trait GWAS[23,24], variant clustering with $\rho_{ij}$ and variance of local gEBV summed over 34 traits). Also, we select variants that had consistent rankings based on their functionality and multi-trait associations between two sexes within each chromosome segment. This leads to the selected informative variants being evenly distributed across the genome, instead of being clustered in some parts of the genome.

In the first step of the method, we used functional information (represented by the FAETH score of sequence variants) and a multi-trait meta-analysis of single-trait GWAS to select 1.7 M

variants out of 17 M. The most significant variants from a GWAS are not necessarily causal[10,35], and in fact may track 2 or more QTL without being in complete LD with any of them. Therefore, we prefer to fine-map and make a further selection of variants when all are fitted simultaneously in our Bayesian method. However, this would be too computationally demanding, so we compromise by first using GWAS and the FAETH score to select the top 10% of variants. That is, we assumed that causal variants and their LD mates will be among the 10% of most significant sequence variants. This selection is expected to represent the majority of informative variants, as both the multi-trait effects[11] and the FAETH score are based on robust methods and relatively large functional datasets with validations. However, it is likely we missed some informative variants due to the datasets used for this study.

In step 2, we clustered variants that had a similar pattern of effects across traits and were in LD. By selecting some variants from every cluster, we attempted to prevent QTL of small effect

from being neglected in the selection of variants to include in the Bayesian analysis.

In step 3, a set of 165k variants were fitted simultaneously in a Bayesian analysis. However, if the Bayesian analysis was based on variants that had been selected using a GWAS from the same data, the estimated effects of these variants would likely be biased. This potential bias was overcome by performing the Bayesian analysis in the cows (bulls) using the pre-selected variants from bulls (cows) GWAS. BayesR assumes that variant effects follow a mixture of four normal distributions with increasing variance. The preselection of variants by GWAS causes bias in the BayesR estimated mixing proportions if the same data is used. Therefore, another way we overcame the bias was to analyse the same data in the BayesR as in the GWAS but using the Dirichlet prior (α) for the distribution of variant effects from another dataset.

In step 4, a set of 80k variants was selected from the 165k used in the Bayesian analysis. For a set of causal or highly informative markers to well predict complex traits across populations/breeds, they need to cover the whole genome and include small-effect variants. Our genome-wide fine-mapping was designed to meet these needs by prioritising variants region-by-region, including clusters (selecting the top 50%) and 50 kb segments for local gEBV calculation (selecting the top 3). Such 'localised' mapping maximised the chance of finding potential causal variants representing genome-wide QTL (Fig. 4), based on the assumption that causal variants, or markers in strong LD with them, existed in these genomic regions. This is supported by our conditional analysis of top variants from selected clusters (Fig. 2) and weighted correlation analysis (Fig. 3) where on average 1 QTL affecting up to 34 traits exist in each 50 kb segment across the cattle genome.

If only one causal variant exists in a genomic segment, we expect animals to rank in the same order for different traits based on their local gEBVs. This proposition was tested by the weighted correlation of local gEBVs (Supplementary Note 2). The variance of local gEBVs for one trait in one sex is inflated by error variance. Therefore, we used the covariance between local gEBVs for trait $i$ and $j$ calculated from two analyses, e.g., (1) local gEBV of trait $i$ based on variants pre-selected in bulls and the prediction equation trained in cows, and (2) local gEBV of trait $j$ based on variants pre-selected in bulls and the prediction equation trained in bulls with Dirichlet α from cows (Supplementary Note 1).

The variance of local gEBV across individuals was used to prioritise the final set of markers. This metric has previously shown power for mapping[36] and is consistent with our observation in the current study (Fig. 4). Although BayesR fits all markers simultaneously to generate marker effects and PP, in some regions where markers are in strong LD no single marker has a high PP (e.g., Fig. 4b). Therefore, we calculated the correlation between variant genotype and the local genetic merit (gEBV) and summed these across traits weighted by the variance of local gEBV for each trait (e.g., Fig. 4e). This way, we maximise the power from the BayesR modelling to locate a small set of the best possible variants.

Our analysis successfully translates big genomic data into a product, the customised XT-50K array, for the global dairy breeding industry. A key feature of this customised array is that most of the markers are supported by either multi-trait association statistics or biological functions (Table 1). Thus, many of them can be putatively causal or at least highly informative. We also included INDELs on the customised array that may play important roles in shaping mammalian complex traits[37]. Compared to the Standard-50K array, the XT-50K array increased genomic prediction accuracies in most scenarios. The observation that the prediction performance of the XT-50K was almost as good as the combined markers from two panels (Fig. 5a) also

supports the conclusion that the XT-50K panel is highly enriched with informative markers. Large improvements of prediction accuracy with the XT-50K panel in small breeds where prediction accuracy is usually low[7,17], also supports the conclusion that across-breed informative markers were captured. Specifically, compared to the Standard-50K panel, there was 11% relative increase in accuracy using Australian multi-breed training dataset to predict New Zealand Jersey traits; and 60%, 30% and 90% relative increase in accuracy using the USA Holstein (GWAS equation instead of BayesR equation) training dataset to predict traits for Australian Jerseys and Reds and New Zealand Jerseys, respectively (Fig. 5a, b). Predictions into Holstein–Jersey mixed breeds also benefited from the XT-50K panel. As the USA single-breed GWAS excluded intergenic and intronic variants which may contain many regulatory variants[28,38], some loss of power is expected for the USA GWAS effects to predict Australasia multi-breed cow traits using the XT-50K panel (Fig. 5b). When using the US GWAS effects to predict the Australian and New Zealand cows, the overall prediction accuracy for NZ traits is lower than that for the AU traits. This may be due to the more distant genetic relationship between NZ and US animals than the relationship between Australian animals and US animals. The difference in the scale of phenotype between the NZ and US animals may also be larger than the difference between AU and US animals.

When using the US Holstein GWAS to predict, the increase in the accuracy of XT-50K compared to Standard-50K panel in the Holstein cattle is smaller than such increase in accuracy in Jersey and Australian Red cattle (Fig. 5b). The fact that there was no Jersey, Australian Red or mixed breeds in the US GWAS dataset limited its power in predicting traits of these two breeds in the Australian and New Zealand cows. This resulted in a lower accuracy for the Standard-50K to predict Jersey and Australian Red traits using the US Holstein GWAS, compared to the accuracy of predicting Holstein traits using the US Holstein GWAS. This would leave more room for the accuracy to improve when using US Holstein GWAS to predict Jersey and Australian Red traits with the XT-50K panel. Overall, the prediction accuracies using the GWAS summary statistics (Fig. 5b) were much more variable than the accuracies using the raw data with BayesR model (Fig. 5a). The use of GWAS summary statistics in genomic prediction (i.e., PRS in humans[15]) of cattle complex traits requires more investigation in the future. Nevertheless, even though only around half of the traits overlapped between the Australian and the US data, putative pleiotropic variants affecting up to 36 US traits were strongly enriched in the Australian XT-50K panel but barely so for the Standard-50K panel (Fig. 5c).

With the increasing amount of phenotypic and omics data available, the precision of the genome-wide fine-mapping will increase, allowing more causal variants to be discovered and customised on the genotyping panel. After several iterations between the fine-mapping and customised genotyping in the future, we expect a chip panel to be highly enriched with causal variants which may be used in combination with single-step genomic prediction methods[21] to allow more accurate genomic selection for many traits.

In conclusion, we present an innovative fine-mapping approach that captures important genome-wide QTL and prioritises informative markers allowing for robust genomic prediction. Our methodology provides additional concepts for fine-mapping with functional and pleiotropic information in general and can be easily applied to other species. Finely mapped markers are used to customise a genotyping array to improve on the existing one and genomic prediction of multiple traits across populations and countries. Overall, we demonstrate a successful case of translating big genomic data into an industry-usable product.

## Methods

**Bull and cow data for variant discovery, genome-wide fine-mapping**. The data analysed in this study was collected by DataGene Australia (http://www.datagene.com.au/) and no further live animal experimentation was required for our analyses. A set of 11,923 bulls and 32,347 Australian cows were used for the main study and this data set has been previously described[7]. Briefly, phenotypes for Australian bulls ($N = 5354$) were obtained from DataGene as daughter trait deviations: i.e. the average trait deviations of a bull's daughters pre-corrected for known fixed effects by DataGene, Australia for the official release of National bull breeding values. Phenotypes for the non-Australian bulls ($N = 6,569$) were derived from their Interbull MACE breeding values (https://interbull.org/ib/interbullactivities) deregressed on the Australian scale and converted to the scale of the daughter trait deviation. Only those bulls' phenotypes which were based on records from more than 15 daughters were included. Cow phenotypes were also processed and corrected for known fixed effects by DataGene. Animal breed origins included Holstein (9739 ♂/22,899 ♀), Jersey (2059 ♂/6174 ♀), mixed breed (0 ♂/2850 ♀) and Australian Red breeds (125 ♂/424 ♀). The 34 traits were ordered by their number of non-missing records and decorrelated by Cholesky transformation[23,24] separately in each sex, so that the traits had minimal phenotypic correlations with each other. After transformation, the $K$th trait can be interpreted as the $K$th original trait corrected for the preceding $K-1$ traits and each Cholesky trait had a variance of close to 1 (Supplementary Table 1). The Cholesky transformation allowed us to use all traits with a varying number of missing values and make phenotypes uncorrelated[23,24].

Most bulls were genotyped with a medium-density SNP array (50 K) or a high-density SNP array (HD: BovineHD BeadChip, Illumina Inc.) while most cows were genotyped with a low-density panel of which approximately 6.9k SNPs overlap with the Standard-50K panel (BovineSNP50 beadchip, Illumina Inc.). The low-density genotypes were first imputed to the Standard-50K panel and then all 50K genotypes were imputed to the HD panel using Fimpute v2.2[7,39]. Finally, all HD genotypes were imputed to whole-genome sequence using Minimac3 with Eagle (v2) to pre-phase genotypes[40,41]. The sequence reference set for imputation was Run 6 of the 1000 Bull Genomes Project[4,34]. All imputed sequence variants with a Minimac3 R$^2$ imputation score < 0.4 were removed. Variants with minor allele frequency (MAF) >0.001 were used. All these sequence variants had a Functional-And-Evolutionary Trait Heritability (FAETH) score where a higher value indicates their stronger functional and evolutionary significance[7]. The genome-wide fine-mapping in our study is described in the following five major steps:

**Step 1: sequence variant prioritisation by pleiotropy and functionality**. This step used two metrics to rank over 17.7 M sequence variants: (1) $p$-value from the multi-trait meta-analysis of GWAS of 34 decorrelated traits and (2) their FAETH score. These variants were ranked based on their multi-trait $p$-value adjusted (divided) by their FAETH score and the variants within the top 10% of such ranking in each sex were prioritised. The prioritised sets of over 1.7 million variants in bulls and cows were LD pruned using Plink 1.9[25] where one of each pair of variants within 5000 kb windows were removed if LD $r^2 > 0.95$.

The multi-trait $p$-value was based on the meta-analysis of 34 single-trait GWAS using the multi-trait $\chi^2$ statistic for variant$_i$:

$$\chi_i^2 = \mathbf{t}_i' \mathbf{V}^{-1} \mathbf{t}_i, \tag{4}$$

published in ref. [11]. $\mathbf{t}_i$ was a $K$ (number of traits = 34) × 1 vector of the signed $t$-values of variant$_i$ effects, i.e., beta/se, for the $K$ traits; $\mathbf{t}_i'$ was a transpose of vector $\mathbf{t}_i(1 \times K)$; and $\mathbf{V}^{-1}$ was an inverse of the $K \times K$ correlation matrix where the correlation was calculated over the all estimated variant effects (signed $t$-values) of the two traits. The $\chi^2$ value of each variant was examined for significance based on a $\chi^2$ distribution with $K$ degrees of freedom to test against the null hypothesis that the variant had no significant effects on any one of the $K$ traits.

The single-trait GWAS of 34 decorrelated traits in each sex fitted a linear mixed model implemented in GCTA[42]:

$$\mathbf{y} = \text{mean} + \textbf{breed}_i + \mathbf{bx} + \mathbf{a} + \text{error} \tag{5}$$

where $\mathbf{y}$ = vector of phenotypes for bulls or cows, $\textbf{breed}_i$ = three breeds for bulls (Holstein, Jersey and Australian Red) and four breed groups for cows (Holstein, Jersey, Australian Red and mixed); $\mathbf{bx}$ = regression coefficient $\mathbf{b}$ on variant genotypes $\mathbf{x}$; $\mathbf{a}$ = polygenic random effects $\sim N(0, G\sigma_g^2)$ where $G$ = genomic relatedness matrix based on all variants. The same GWAS model as above, but without including variants in the model ($\mathbf{bx}$), was applied to estimate variance components for the calculation of decorrelated trait heritability.

**Step 2: Variant prioritisation within sliding-window clusters**. This step was based on the metric $\rho_{ij} = r_{\text{cor}}(t_{\text{variant}_i}, t_{\text{variant}_j}) \times r_{\text{LD}}(\text{variant}_i, \text{variant}_j)$ (1) within each sex. The motivation behind $\rho_{ij}$ was that the combined measure of the genotype correlation and effect correlation between variant pairs was expected to capture sequence variants tagging the same QTL. The reason for choosing 5 Mb as the size of the sliding-window was that it was likely to be large enough to see the differentiation of distributions of LD between sequence variants and to potentially include several QTL. $\rho$ was computed for all variant pairs within 5 Mb sliding-windows (step was 2.5 Mb). Within each window, variant pair $\rho$ values were

clustered using graph-based Random Walks determining densely connected sub-graphs (clusters)[43] implemented in igraph[44]. Up to 50 clusters were retrieved in each window and in total, 30,514 clusters were retrieved in bulls and 30,440 were retrieved in cows. There were 1–98 (average 11.9) variants per cluster in bulls and 1–92 (average 11.7) variants per cluster in cows. The Random Walks algorithm was chosen because of its clustering capability for a dense network with high computational efficiency[43]. The Random Walks algorithm can be operated at an agglomerative mode which learns the hierarchy of a network, and this was similar to the conventional hierarchical cluster algorithms which tend to be much less efficient. The hierarchical tree for each sliding-window was cut to form individual clusters by assuming that up to 50 clusters could be reasonably formed to represent QTL within each 5 Mb window. Within each cluster, unique variant members were determined and ranked based on their multi-trait $p$-value adjusted by the FAETH score. Then variants ranked within the top 50% (inclusive) were chosen as the prioritized variants. The R codes of variant clustering with test datasets are available at https://github.com/rxiangr/SNP_cluster_ranking.

To verify if those clusters represented different pleiotropic QTL, a conditional GWAS was conducted using GCTA-COJO[45]. Thirty-four single-trait GWAS were re-run conditioned on variants ranked within top 100 (out of the 165k variants after prioritisation within sliding-window clusters) for their adjusted $p$-value by the FAETH score in both sexes. The meta-analysis of the conditional GWAS also used Eq. (4).

The reason for fitting variants representing certain clusters, instead of fitting variants representing all clusters, in the conditional GWAS was that, in theory, if identified clusters represented different pleiotropic QTL, fitting the top variants from these clusters in the conditional GWAS would only reduce the significance of the effects of the variants from these selected clusters, not the significance of the effects of the variants from other clusters.

**Step 3: BayesRC mixture modelling across sexes**. The across-sex design was used to maximise the power from separated male and female cattle and to reduce bias as much as possible by not training and predicting using the same population. Those 165k variants prioritised from bulls as described above were trained by BayesRC in cows. Accordingly, those 165k variants prioritised from cows were trained by BayesRC in bulls. The BayesRC algorithm[6] added a feature to the BayesR algorithm[19] of including a priori independent biological information to allocate each variant to a specific category ('c'). Similar to BayesR, BayesRC modelled the variant effects as mixture distribution of four normal distributions including a null distribution, $N(0, 0.0\sigma_g^2)$, and three others: $N(0, 0.0001\sigma_g^2)$, $N(0, 0.001\sigma_g^2)$ and $N(0, 0.01\sigma_g^2)$, where $\sigma_g^2$ was the additive genetic variance for the trait. This mixture of distributions is modelled independently in each category of variants to allow for different mixture models per category. The starting value of $\sigma_g^2$ for each trait within each sex was determined by GREML implemented in the MTG2 software[46] using a single genomic relationship matrix made of all 17 M sequence variants. For each trait, the BayesRC model fitted in the training dataset was:

$$\mathbf{y} = \mathbf{Xb} + \mathbf{Wv} + \mathbf{e} \tag{6}$$

$\mathbf{y}$ was the vector of each decorrelated trait; $\mathbf{X}$ was the design matrix allocating phenotypes to fixed effects; $\mathbf{b}$ was the vector of fixed effects, such as breeds; $\mathbf{W}$ was the design matrix of marker genotypes; centred and standardised to have a unit variance; $\mathbf{v}$ was the vector of variant effects, distributed as a mixture of the four distributions (described above); $\mathbf{e}$ = vector of residual errors.

The C component of BayesRC had three categories of $c$ ($c = 3$): variants that were ranked 'high' (top 1/3), medium (middle 1/3) and low (bottom 1/3) for their FAETH score. Within each category $c$, an uninformative Dirichlet prior ($\alpha$) was used for the proportion of effects in each of the four normal distributions of variant effects: $P_c \sim Dir(\alpha_c)$, where $a_c = [1, 1, 1, 1]$. $a_c$ was updated each iteration within each category: $P_c \sim Dir(\alpha_c + \beta_c)$, where $\beta_c$ was the current number of variants in each of the four distributions within category $c$, as estimated from the data.

After each single-trait BayesRC run, the posterior probability (PP) of having a non-zero effect was obtained for each variant by summing the proportion of iterations the variant was allocated to each of the 3 mixture distributions with non-zero variance[6]. Across 34 single-trait BayesRC, each variant had $PP_i$ where $i \sim 1:K$ ($K = 34$ in this study). Then, a multi-trait PP of each variant to have a non-zero effect for all traits was calculated as

$$\pi = 1 - \prod_1^K (1 - PP_i). \tag{7}$$

The local gEBV was calculated using the conventional gEBV methods for each trait (e.g., ref. [26]), except that the variants used to calculate the local gEBV were from each 50 kb segment:

$$\hat{y}_{v_l} = W_{l_1:l_n} \hat{v}_{l_1:l_n} \tag{8}$$

where $\hat{y}_{v_l}$ was the local gEBV of the segment $l$, $W_{l_1:l_n}$ was the design matrix of marker genotypes (from Eq. (6)) for variant 1 to variant $n$ within the segment $l$, and $\hat{v}_{l_1:l_n}$ was the variant effects from the training dataset. The use of local gEBV took advantage of the BayesR model where variants were analysed jointly which accounted for LD. The variance of local gEBV can be used to prioritise informative

chromosome segments as shown by our previous study[26]. The local gEBV was calculated using each segment for each one of the 34 traits within each sex.

**BayesR mapping.** In the above described BayesRC modelling scenario, variants prioritised from one sex were trained in the opposite sex to avoid bias. However, it was also possible for variants prioritised from one sex were trained in the same sex, but the prior of the variant effect distribution (i.e., the Dirichlet prior $\alpha$) was determined by the variant effect distribution (mixing proportion) previously estimated from the opposite sex with BayesRC mapping. According to the implementation of BayesR[19], in each iteration, the proportions of variant effects in each one of the four effect distributions was updated as: $P \sim Dir(\alpha + \beta)$[19] where $\beta$ was the current number of variants in each of the four distributions. This meant that a predetermined Dirichlet prior $\alpha$ informed by the mixing proportions estimated in the opposite sex with BayesRC will influence each iteration of the BayesR learning, i.e., the mixing proportion of the current BayesR modelling will lead towards the previous mixing proportion of the BayesRC modelling.

Using the above-described design, those 165k variants prioritised from bulls were trained by BayesR in bulls. Accordingly, those 165k variants prioritised from cows were trained by BayesR in cows. For training with BayesR in each sex, instead of using the uninformative Dirichlet prior $\alpha$ of [1,1,1,1] as the starting value, an informative Dirichlet prior $\alpha$ of $[\xi_1,\xi_2,\xi_3,\xi_4]$ was used to initiate BayesR training. $\xi_1$ was the number of variants falling into the null distribution, $N(0, 0.0\sigma_g^2)$, in the previous BayesR training for the same trait in the opposite sex, $\xi_2$ was the number of variants falling into the 2nd distribution of $N(0, 0.0001\sigma_g^2)$, $\xi_3$ was the number of variants falling into the third distribution of $N(0, 0.001\sigma_g^2)$ and $\xi_4$ was the number of variants falling into the fourth distribution of $N(0, 0.01\sigma_g^2)$. $\sigma_g^2$ was the additive genetic variance for the trait. For each one of the 34 traits, the BayesR training in the same sex where the 165k variants were prioritised (discovered) used Eq. (6). After the genomic predictors were trained, Eq. (8) was used to calculate the local gEBV with the genotype data from the opposite sex. A more practical explanation of such across-sex BayesR implementation is detailed in Supplementary Note 1.

**The number of QTL per 50 kb segment by analysis of (co)variance of local gEBV.** As described above, using variants from each segment, two sets of local gEBV were estimated for $K$ traits ($K = 34$ in this study) within each sex: one set was generated from the BayesRC scenario where variants were prioritised (discovered) in one sex (e.g., bull), trained in the opposite sex (e.g., cow) and estimated the local gEBV in that sex (e.g., cow). The other set was generated from the BayesRC scenario where variants were prioritised (discovered) in one sex (e.g., bull), trained in the same sex (e.g., bull) and estimated the local gEBV in the opposite sex (e.g., cow). Therefore, for each segment, a $K \times K$ asymmetric variance-covariance matrix can be built (Fig. 3a). This was used to calculate the weighted correlation for each segment and its inverse was used to infer the number of QTL within each segment. The weighted correlation was calculated as

$$r_{\text{weighted}} = \frac{\sum_i^n \sum_{j(i \neq j)}^n |C_{ij}|}{\sum_i^n \sum_{j(i \neq j)}^n \sqrt{V_{ii} \times V_{jj}}}, \tag{9}$$

where $\sum_i^n \sum_{j(i \neq j)}^n |C_{ij}|$ was the sum of the approximate absolute values of the off-diagonal elements of the matrix and $\sum_i^n \sum_{j(i \neq j)}^n \sqrt{V_{ii} \times V_{jj}}$ was the sum of the pairwise geometric mean of the self-exclusive diagonal elements. For the weighted correlation calculation using the per segment matrix, only the diagonal elements with positive values and their associated rows and columns in the asymmetric matrix were considered. Also, to properly account for the impact of negative off-diagonal values on the sum, we used an 'approximate absolute value' approach and this was implemented as a programmatic sign-flipping process for the asymmetric matrix for each segment (Supplementary Note 2).

**Step 4: Causal marker prioritisation by partitioning local gEBV variance to variants.** The variance of local gEBV estimated with BayesRC scenario across individuals was calculated for each segment for each trait within each sex. Then, Eq. (3), $\text{Var}_{g_{\text{local}}}(\text{variant}) = \text{Var}(g_{\text{local}}) \times r^2(\mathbf{g}_{\text{local}}, \mathbf{x})$ was used to partition the local gEBV variance, $\text{Var}(g_{\text{local}})$, to each variant, based on the squared correlation ($r^2$) between the value of local gEBV ($g_{\text{local}}$) and the genotype of the variants ($\mathbf{x}$) within the segment where the local gEBV was estimated. The use of the squared correlation was to prevent the negative value of the $\text{Var}_{g_{\text{local}}}(\text{variant})$ due to the negative $r$ which would reduce the sum of $\text{Var}_{g_{\text{local}}}(\text{variant})$. $\text{Var}_{g_{\text{local}}}(\text{variant})$ of each marker that entered the BayesRC mapping was estimated for each trait and added up across 34 traits, i.e., $\sum_1^{34} \text{Var}_{g_{\text{local}}}(\text{variant})$, within each sex. This allowed the ranking of the analysed variants based on $\sum_1^{34} \text{Var}_{g_{\text{local}}}(\text{variant})$ within each sex and variants within the top 3 of such ranking for a given local gEBV window in both sexes were selected to make up the 80k variants (83,455) as shown in Supplementary Fig. 11. The R codes for analysing the correlation between genotypes and local gEBV with its test datasets are available at https://github.com/rxiangr/SNP_correlation_local_gEBV_variance.

**Step 5: Infinium XT beadchip design.** The final goal of this study was to use the selected list of 80k markers to design a custom XT genotyping panel with up to 50k variants for the Dairy industry (XT-50K). In this instance, the selected variants were to be added to an existing XT array of ~9000 variants. The steps for the design of the Infinium XT-50K (Illumina Inc) beadchip are detailed in Supplementary Fig. 12. Briefly, flanking sequence for each SNP and INDEL was extracted ($+/-150$ bases) and any other known variants within the flanking sequence were identified, using the variant list from Run 6 of the 1000 Bull Genomes Project (REFs), and then masked (replaced with N). All the INDELs and SNPs were processed using DesignStudio (Illumina Inc., https://sapac.illumina.com/informatics/sample-experiment-management/custom-assay-design.html) according to the manufacturer's instructions. Only designable markers with a design score >0.4 were kept. Where possible Infinium I design (requiring 2 probes), were swapped with designable Infinium II (requiring 1 probe) mates with LD $r^2 > 0.9$ in the 80k list and design score >0.4.

**Validation using additional Australian, New Zealand and US cattle data.** Phenotypes from an additional 28.2k Australian cows with no overlap with the above described 44k bulls and cows were provided by Datagene and processed in the same way as described above for the main set of cows. The new Australian cows consisted of 24.4 Holsteins, 2.5k Jerseys and 1.2k Australian Reds. The raw phenotype data of an additional 21.2k New Zealand cows were obtained from DairyNZ (https://www.dairynz.co.nz/) and were merged into the Australian cow database by DataGene, enabling the new Australian cows and the NZ cow data to be processed jointly by DataGene with all the national Australian data to produce deregressed proof phenotypes (phenotypes corrected for herd-year-season-age) for milk, fat and protein yields[47]. The New Zealand cows consisted of 8.6k pure Holsteins, 3.9k pure Jerseys and 8.7k mixed breed of Holstein and Jersey.

The genotype data of the new Australian and New Zealand cows were imputed sequence variants of the XT-50K and Standard-50K markers with the same methods as described above. The breed codes of the new Australian and New Zealand cows were cross-checked using a principal component analysis of a GRM based on 8.5k low-density genotypes. Two separate GRMs were constructed for the Australian and the New Zealand cows and each included purebred Holstein and Jersey bulls from the same country. Based on both pedigree and PCA, cows were allocated a breed code indicating their approximate breed proportion. Crossbred cows in the new Australian population were excluded due to very small numbers.

In the first validation test, three sets of markers, (1) XT-50K markers, (2) standard-50K markers and (3) the combined markers from (1) and (2) were trained by BayesR in the new Australian cows and predicted into the New Zealand cows. This training used the same model as Eq. (6) and the prediction of gEBV used the same model as Eq. (8), except that instead of using markers from chromosome segments, all markers were used to predict the gEBV of 3 traits. The Pearson correlation $r$ between gEBV and the individual phenotype within each breed of the validation cows (the same for all validation analysis) was used as the proxy for the prediction accuracy.

In the second validation test, we used published single-trait GWAS summary statistics for 36 traits of 27.1k US Holstein bulls[29]. The summary statistics included the beta, $p$-values and coordinates of ~2.7 million sequence variants that were imputed from Run 5 of the 1000 Bull Genomes Project[4,34] with the imputation cut-off set by the authors[29]. The authors[29] also excluded those variants annotated as intergenic and intronic variants by Ensembl VEP[31]. These 2.7 M US variants had an overlap of 2.3 M variants with the 17.7 M Australian variants, an overlap of 36.2k with the standard-50K markers, and an overlap of 27.7k with the 46.5k Australian XT-50K markers.

The estimated variant effects from the US GWAS that overlapped with the Australian XT-50K and standard 50 K markers were used to predict (similar to Eq. (8)) milk yield, fat yield and protein yield in the new Australian and New Zealand cows. Because the Standard-50K markers found in the US GWAS were over 30% more than the XT-50K markers found in the US GWAS (36.2k VS 27.7k), a random set of the Standard-50K markers presented in US GWAS were dropped to leave the total number used at 27.7k to match the number of variants on the Australian XT-50K panel. The GWAS effects of these 27.7k markers were used to calculate the EBV in the new Australian and New Zealand cows using MTG2[46] with the option of -sbv b.

In the third validation test, we conducted a multi-trait meta-analysis of 36 US single-trait GWAS (Supplementary Table 3 and ref. [29]) with the method of Bolormaa et al.[11]. This analysis generated a multi-value $p$-value for each variant presented in the US GWAS, which was tested against the null hypothesis that each variant had no effect on any one of the 36 US traits. This meant that a variant with small a multi-trait $p$-value would be associated with many traits. Then, for 100 thresholds of the multi-trait $p$-value, the significance of the enrichment of significant pleiotropic variants within the XT-50K markers and Standard-50K markers were tested. At each multi-trait $p$-value threshold, e.g., $p = 0.05$, the following counts of variants were obtained: $n_{XT\&sig}$: the number of variants with multi-trait $p < 0.05$ and were also the XT-50K markers and $n_{STD\&sig}$: the number of variants with multi-trait $p < 0.05$ and were also the Standard-50K markers. Other constant numbers counted included: $n_{XT}$: the total number of XT-50K markers presented in the US GWAS; $n_{STD}$: the total number of Standard-50K markers presented in the US GWAS; $n_{sig}$: the total number of variants with multi-trait

$p < 0.05$; and $n_u$: the total number of unique variants presented in the XT-50K, Standard-50K and US GWAS. In R (v3.5.1), the hypergeometric test was conducted for these numbers at each multi-trait p threshold as implemented in phyper; for XT-50K markers: $phyper(n_{XT\&sig} - 1, n_{XT}, n_u - n_{XT}, n_{sig}, \text{lower.tail} = F)$ and for Standard-50K markers: $phyper(n_{STD\&sig} - 1, n_{STD}, n_u - n_{STD}, n_{sig}, \text{lower.tail} = F)$.

**Reporting summary**. Further information on research design is available in the Nature Research Reporting Summary linked to this article.

## Data availability

The information regarding the XT-50K markers and the multi-trait GWAS summary statistics of the 46.5k markers of Australian bulls, cows and the US bulls are accessible via figshare at https://melbourne.figshare.com/articles/dataset/Marker_information_of_the_XT-50K_panel/13523837 with the https://doi.org/10.26188/13523837. The functional and evolutionary data are publicly available via the FAETH score at https://melbourne.figshare.com/articles/The_Functional_And_Evolutionary_Trait_Heritability_FAETH_score_of_over_17_million_cattle_sequence_variants/7660277/2. Additional GWAS results for Australian animals can be found at Xiang et al.[24]. DataGene Australia (http://www.datagene.com.au/) are custodians of the raw phenotype and genotype data of Australian farm animals. DairyNZ (https://www.dairynz.co.nz/) are custodians of the raw phenotype data of New Zealand farm animals and CRV (https://www.crv4all-international.com/) are custodians of the raw genotypes of New Zealand farm animals. Details of the data access to the GWAS results from US animals used for validation of XT-50K markers can be found at Jiang et al.[29]. The DNA sequence data as part of the 1000 bull genome project[4,34,48] is included in NCBI BioProjects PRJNA431934, PRJNA238491, PRJDB2660, PRJEB18113, PRJEB1829, PRJEB27309, PRJEB28191, PRJEB9343, PRJNA210519, PRJNA210521, PRJNA210523, PRJNA279385, PRJNA294709, PRJNA316122, PRJNA474946, PRJNA477833, PRJNA494431, PRJDA48395, PRJNA431934, PRJNA238491. Other supporting data are shown in the Supplementary Information of the current manuscript.

## Code availability

The GWAS used plink (https://www.cog-genomics.org/plink/) and GCTA (https://cnsgenomics.com/software/gcta/#Overview). The multi-trait meta-analysis implemented in R is published at https://melbourne.figshare.com/articles/Effect_Direction_MEta-analysis_EDME_of_GWAS/11730939/1. Variant annotation used public Variant Effect Prediction of Ensembl and NGS-SNP (http://stothard.afns.ualberta.ca/downloads/NGS-SNP/). The details of BayesRC can be found at MacLeod et al.[6]. The implementation of weighted correlation analysis is detailed in Supplementary Note 2. The R codes of variant clustering with test datasets are available at https://github.com/rxiangr/SNP_cluster_ranking and the R codes of correlation between genotypes and local gEBV with its test datasets are available at https://github.com/rxiangr/SNP_correlation_local_gEBV_variance.

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

## Acknowledgements

Australian Research Council's Discovery Projects (DP160101056 and DP200100499) supported R.X. and M.E.G. DairyBio, a joint venture project between Agriculture Victoria (Melbourne, Australia), Dairy Australia (Melbourne, Australia) and the Gardiner Foundation (Melbourne, Australia), funded computing resources used in the analysis. The authors also thank the University of Melbourne, Australia for supporting this research. No funding bodies participated in the design of the study nor analysis, or interpretation of data nor in writing the manuscript. DataGene and CRV provided access to the reference data used in this study. DairyNZ provided access to the validation data used in this study. We thank Gert Nieuwhof, Kon Konstantinov and Timothy P. Hancock (DataGene) and staff from DairyNZ for preparation and provision of data. We thank Dr. Mekonnen Haile-Mariam for deriving the deregressed phenotypes from international MACE and Dr. Sunduimijid Bolormaa for sequence variant data imputation. We thank Dr. Majid Khansefid for the curation of validation datasets.

## Author contributions

M.E.G. and R.X. conceived the study. R.X. and I.M.M. implemented the design and analysis. I.M.M., H.D.D., C.S., G.d.J. and E.O. provided data and assisted with study design. R.X., I.M.M. and M.E.G. analysed data. R.X., I.M.M. and A.J.C. designed the customised XT-50K genotyping array. R.X. and M.E.G. wrote the paper. R.X., M.E.G., I.M.M., H.D.D. and A.J.C. revised the paper. All authors read and approved the final manuscript.

## Competing interests

The authors declare no competing interests.

## Additional information

**Supplementary information** the online version contains supplementary material available at https://doi.org/10.1038/s41467-021-21001-0.

