## [Peer Review File · Nature Communications]

Reviewers' comments:

Reviewer #1 (Remarks to the Author):

This is a lengthy, convoluted and difficult to follow paper. I do not see any major flaw in the sense that the new list of SNPs is better than the current one, but there is a lot of ad hoc reasoning. The reader is overwhelmed by information and tables that is often useless (eg table s1, s2).

Results seem a repetition of methods, which makes it even harder to be read. I would suggest to shorten results without repeating methods, show a small example on fine mapping but better explained and go to the point of crossvalidation. In here, I would ask for comparing different SNP sets in order to assess which was the most influential step. For instance, would the 50k FAET top score SNPs be any better than the final set?

It would be useful to plot SNP frequencies of selected SNPs for each breed as well as genomewide and 50k SNPs.

Additional

How are local gEBVs computed? How does it account for LD?

I92-93: theoretical estimation of the existence of causal QTL? How is this done? I wish I would have that rule.

I148: I agree significance should not drop but for how much? where do you set the threshold.

I172: the variance of local gEBV as proof of QTL is not a standard criterion.

Minor

The reader is not aware of the breeds unless the methods is read, and not much info on the phenotypes is actually provided.

Introduction does not flow

Several misspellings

K for 1000 is k (lower case)

Reviewer #2 (Remarks to the Author):

This paper presents an approach to identify causative SNP from sequence data and then using them for genomic prediction. Most of the techniques cited are from analyses of human data. To tap into prediction for many traits, 34 traits are decomposed into independent factors by Cholesky decomposition. Causative SNPs are identified using GWAS with heuristics plus existing QTL data (FAETH scores), combining the information from all 34 factors. The identification is separate from cows and bulls. On top of it, one comparison uses fraction of SNP present in the sequence-derived SNP from the US. Accuracies are calculated for NZ cattle from AU predictions and for AU and NZ cattle from US-base subset. With the selected variants the accuracies with the selected SNP are always higher than with a standard set of SNP. However, the accuracies are low, below 50%, while commonly the accuracies from genomic predictions are over 0.7 (reliability 0.5 and higher).

The paper draws heavily on references from analysis of human data. However, the prediction of breeding values in farm animals is only possible because of small effective population size (N_e). Subsequently the number of independent chromosome segments is around 5-15k, as opposed to as much as 1 M in human. With large data, when the values of most segments can be predicted very well, the prediction accuracy is high without QTL identification. Also, with larger data, the impact of SNP selection decreases to almost none (e.g., Karaman et al., 2016), and most national

evaluation centers do no longer select SNPs for analyses as gains from such selection are small. Senior authors know this issue very well, as they were authors on papers of genomic accuracies based on a small number of segments. That genomic prediction in farm animals is possible due to small N_e is crucial information for readers not familiar with animal data. Small N_e is also a reason why a 50k or so SNP work so well.

The study uses cow and bull data separately, as if they were different populations. The rationale was that the bulls are preselected. P-values for SNP based on cow and bull data were mostly very different. In fact, cows and bulls form one population, some of the cows' information was probably included in bulls' information as daughter information. Cows carry less information than bulls as bulls' predictions are based possibly on hundreds of daughters. Also, cows may be preselected as well. To maximize accuracy while minimizing biases, the authors could use methods that allow to use all genotyped animals together with ungenotyped animals, e.g., single-step Bayesian Regressions.

The authors use Cholesky decomposed traits. Such decomposition is impossible for 34 traits as the variance covariance matrix is certainly not positive definite, perhaps has a rank of 15, unless the authors "made it" positive definite. But then the results would make little sense for most of the transformed traits. There is an established technique for handling a large number of traits, possibly with a singular variance-covariance structure, called canonical transformation.

Are accuracies in Figure 5 for bulls or cows? How are they derived? If for bulls, are they accuracies or reliabilities?

The results are given only for 3 production traits. What about the other traits? It would be interesting to see results in other traits, e.g., fertility and productive life.

There is no legend in Figure 5, graph B. Are they for XT common with the US?

The accuracies for NZ cattle based on US SNP data are much lower than those based on AU data. Why is it so? Is it because the US derived potential causative SNP are much poorer than those derived by AU? With standard 50k SNP, the accuracies for NZ Jerseys and their crosses in A & B should be the same and they are different.

The paper uses many potential causative SNP. Based on experiences, most of these SNP have an effect too small to be well estimated. Can some accuracies be derived using only 10, 100 or 1000 variants with the biggest effect?

Reviewer #3 (Remarks to the Author):

This paper outlines how methods to fine-map genome-wide informative sequence variants with pleiotropic effects and functional significance were employed to select more informative markers for a 50K SNP chip, and the resulting SNP chip improved the accuracy of genomic predictions in dairy cattle relative to the Standard-50K panel (BovineSNP50 beadchip, Illumina Inc). That is good news, but not entirely unexpected given the old Standard-50K panel was designed over a decade ago with basically no biological function data. But having said that the current analysis is a quite complete analysis using up-to-date data and biological information. It is limited to dairy evaluations and breeds though, or at least of the data presented.

One major concern I have is that the authors just compared their XT-50K panel to the standard the Standard-50K panel (BovineSNP50 beadchip, Illumina Inc), and nothing else. The approach to generate the XT-50K is relatively complicated. It not clear how the XT-50K panel performance would compare to simpler methods? For example, just using the significant SNPs from BayesRC

and some markers evenly distributed along the genome.

I was also surprised to see no reference to the other bovine chips like 800K HD and GGP-F250? <https://link.springer.com/article/10.1186/s12711-019-0519-x> . This paper was not even referenced in the citations. At least a discussion comparing how this product (https://genomics.neogen.com/pdf/slicks/ag229_ggp_f250_array_flyer.pdf) was developed and how the methods used to design the XT-50K panel were different/better would have been appropriate. Likewise how does the XT-50K compare to the improvement in accuracy from 800K HD?

The introduction had a number of statements about phenotype prediction and improved mapping with the XT-50K which is not what is contained in the paper. The authors need to define what they mean by precision breeding. This needs significant rewriting. The discussion and conclusion likewise need to reflect the data in the paper and not overstate the findings/outcome of this paper. Specific comments are in the attachment.

We thank the reviewers who have spent time and effort providing comments and suggestions on our manuscript. Please find our point-to-point response to reviewers' comments in blue text in what follows.

Reviewer #1 (Remarks to the Author):

This is a lengthy, convoluted and difficult to follow paper. I do not see any major flaw in the sense that the new list of SNPs is better than the current one, but there is a lot of adhoc reasoning. The reader is overwhelmed by information and tables that is often useless (eg table s1, s2).

Author response: We thank the reviewer for spending time reviewing our manuscript and provide positive comments on our methodology. We apologize for the many necessary technical details that are included in the Supplementary files. While these might have reduced the readability of the manuscript we have retained them because the reviewer wanted more details about the phenotypes which is included in Table S1. As regards the length of the paper, the main text of the initial submission has around 9500 words excluding references and less than 6000 words excluding methods. We believe such length of a manuscript describing a complex study that has integrated several different data types and has analysed over 30 traits with a total sample size > 100,000 animals from three countries may be reasonable. In order to increase the readability of the manuscript, we have significantly revised the manuscript throughout, especially we have re-written the introduction. We have also shortened the results section as requested by the reviewer. However, because more results are requested by the reviewer and other reviewers, we had to add some minimal descriptions into the existing manuscript. We hope these efforts increase the readability of the manuscript.

Results seem a repetition of methods, which makes it even harder to be read. I would suggest to shorten results without repeating methods, show a small example on fine mapping better explained and go to the point of crossvalidation. In here, I would ask for comparing different SNP sets in order to assess which was the most influential step. For instance, would the 50k FAET top score SNPs be any better than the final set?

Author response: to meet the requests of the reviewer, we have reduced the amount of technical details in the results as requested by the reviewer. Although, we did not feel that the initial results section was too long as the results plus figure legends have a length of fewer than 4000 words.

As requested by the reviewer, we have conducted additional analysis where we trained 8 sets of variants selected differently using data of 6 traits of 28.1k Australian cows and predicted into additional 14.1k Australian cows with three breeds. Tested variant selections included top 10k and 80k variants with FAETH ranking, top 10k and 80k variants with GWAS ranking, top 10k and 80k with BayesRC posterior probability (PP) ranking, and our methods of BayesRC IgEBV and the final selection of XT50K SNPs. These results are shown in the new Supplementary Note S3 and are briefly mentioned in the result section of '*Illustration of mapping and fine mapping of QTL*'. The new results supported our method of selecting variant using local gEBV from BayesRC to be the best. We also found that top variants with FAETH and BayesR PP ranking are the 2nd best, with the top variants selected by GWAS being the worst.

It would be useful to plot SNP frequencies of selected snps for each breed as well as genomewide and 50k snps.

Author response: as requested, we have plotted the minor allele frequency (MAF) of the all SNPs, the standard 50K SNPs and the XT50K SNPs in each breed. These results are shown in Figure S4 and briefly mentioned in the 1st paragraph in the results section of '*Final selection for inclusion on custom Infinium XT-50K beadchip*'. On average, the XT50K variants were as common as the standard50K SNPs in the Holstein breed, and were a slightly rarer than the standard50K SNPs in Jersey and Australian Red. The latter is due to the lower MAF of all sequence variants in these 3 breeds which all have smaller sample size than Holstein. The XT50K SNPs were much commoner than random sequence variants in all breeds.

Additional

How are local gEBVs computed? How does it account for LD?

Author response: the calculation of the local gEBV based on BayesR models has been detailed in both results and methods. For example, 'The BayesRC analysis resulted in a prediction equation that predicted phenotype (breeding value) for each trait from the 165K variant genotypes. This prediction equation was also applied to the variants within each 50kb chromosomal segment to generate local gEBV for 34 traits.' in the 1st paragraph of the section 'Bayesian mixture modelling across sexes' and 'The local gEBV was calculated using the conventional gEBV methods for each trait (e.g., ¹), except that the variants used to calculate the local gEBV were from each 50Kb segment: $\hat{y}_{v_l} = W_{l_1:l_n} \hat{v}_{l_1:l_n}$ (equation 8), where \hat{y}_{v_l} was the local gEBV of the segment l , $W_{l_1:l_n}$ was the design matrix of marker genotypes (from equation 6) for variant 1 to variant n within the segment l , and $\hat{v}_{l_1:l_n}$ was the variant effects from the training dataset.' in the 4th paragraph of the section 'Step 3: BayesRC mixture modelling across sexes.' in methods. Thus, the local EBV is the effect of this 50kb segment of the animal's genome after accounting for the rest of the genome. In this way it accounts for LD. Additional explanations have been added to this method section: 'The use of local gEBV took advantage of the BayesR model where variants were analysed jointly which accounted for LD. The variance of local gEBV can be used to prioritise informative chromosome segments as shown by our previous study'

192-93: theoretical estimation of the existence of causal QTL? How is this done? I wish I would have that rule.

Author response: we have changed 'theoretical estimation of the existence of causal QTL' in both Figure 1 and its legend in this section to 'predicted number of pleiotropic QTL per segment'. The latter is a more accurate description of the work we have done which are detailed in the section '*The number of pleiotropic QTL per segment by analysis of (co)variance of local gEBV*' in results and methods.

1148: I agree significance should not drop but for how much? where do you set the threshold.

Author response: we did not need to set a threshold for the drop in significance because the p-value of the SNPs in clusters not selected for the conditional GWAS actually increased. In Figure 2C we showed the p-value for SNPs from clusters selected for conditional GWAS and for SNPs from clusters not selected for conditional GWAS. The average $-\log_{10}(p)$ for SNPs from clusters not selected for conditional GWAS actually slightly increased.

1172: the variance of local gebv as proof of QTL is not a standard criterion.

Author response: we have revised this sentence. It is now written as 'The variance of local

gEBV across individuals has previously been shown to be a useful metric for prioritising informative genomic regions and was used here'

Minor

The reader is not aware of the breeds unless the methods is read, and not much info on the phenotypes is actually provided.

Author response: we have provide a brief description of breeds used in the 1st paragraph of results: Our genome-wide fine-mapping in Holstein (9,739 ♂ / 22,899 ♀), Jersey (2,059 ♂ / 6,174 ♀), mixed breed (0 ♂ / 2,850 ♀) and Australian Red breeds (125 ♂ / 424 ♀) had five major steps as described in Figure 1.'

Introduction does not flow

Author response: we apologize for the readability of the introduction. We have completely re-written the introduction section in the revised manuscript.

Several misspellings

Author response: we apologize for these errors. We have carefully check the revised manuscript and corrected spelling errors.

K for 1000 is k (lower case)

Author response: we have corrected K to k when describing the number of animals and/or SNPs throughout the manuscript.

Reviewer #2 (Remarks to the Author):

This paper presents an approach to identify causative SNP from sequence data and then using them for genomic prediction. Most of the techniques cited are from analyses of human data. To tap into prediction for many traits, 34 traits are decomposed into independent factors by Cholesky decomposition. Causative SNPs are identified using GWAS with heuristics plus existing QTL data (FAETH scores), combining the information from all 34 factors. The identification is separate from cows and bulls. On top of it, one comparison uses fraction of SNP present in the sequence-derived SNP from the US. Accuracies are calculated for NZ cattle from AU predictions and for AU and NZ cattle from US-base subset. With the selected variants the accuracies with the selected SNP are always higher than with a standard set of SNP. However, the accuracies are low, below 50%, while commonly the accuracies from genomic predictions are over 0.7 (reliability 0.5 and higher).

Author response: we thank the reviewer for spending time considering our manuscript. As the reviewer stated, the main purpose of our study is to demonstrate a method of fine-mapping using large datasets and then to identify a set of informative variants to be included in a customised SNP chip. We use genomic prediction analysis to validate the selected variants to show that they lead to higher prediction accuracy. A likely advantage of using markers close to the causal variants is that the prediction derived from one breed or ethnic group will work in other breeds or ethnic groups. Our results show that our predictions work in different countries and different breeds. Unfortunately, this may make the paper more complex but it is an important objective.

We apologize if our presentation of the results has led to some misunderstanding of accuracy. We assume that the reviewer takes accuracy to mean the correlation between estimated breeding value and the true breeding value. However, in the manuscript, we define the

prediction accuracy as the correlation between the estimate and the cow phenotype. This accuracy is expected to be low for traits of low heritability^{1,2}. As shown in Table S1, the average heritability across 34 cow traits estimated using 17M variants is 0.164 ± 0.03 and none of the cow traits had heritability > 0.7 . We have added sentences to the 1st paragraph of the results where we started to describe genomic prediction accuracy (1st paragraph of ‘*Validation of the new XT-50K custom array for genomic prediction in global cattle data*’) to stress that we have used cow phenotypes to conduct genomic prediction.

The paper draws heavily on references from analysis of human data. However, the prediction of breeding values in farm animals is only possible because of small effective population size (N_e). Subsequently the number of independent chromosome segments is around 5-15k, as opposed to as much as 1 M in human. With large data, when the values of most segments can be predicted very well, the prediction accuracy is high without QTL identification. Also, with larger data, the impact of SNP selection decreases to almost none (e.g., Karaman et al., 2016), and most national evaluation centers do no longer select SNPs for analyses as gains from such selection are small. Senior authors know this issue very well, as they were authors on papers of genomic accuracies based on a small number of segments. That genomic prediction in farm animals is possible due to small N_e is crucial information for readers not familiar with animal data. Small N_e is also a reason why a 50k or so SNP work so well.

Author response: we have cited many references from human genetics because using functional information to find causal or informative SNPs has been studied far more in humans than in animals. This is exactly why our study has been conducted in the first place. We agree with the reviewer about the importance of low N_e in determining accuracy and have changed the manuscript to include this point. While it is true that with a large enough training population the accuracy can always be made high, it is not always possible to achieve a sufficiently large training population, for instance, in small breeds or for traits that are hard to measure. Our objective is a prediction equation which will work in populations other than the one used to train the prediction. In this case the N_e is not small and the number of chromosome segments whose effects must be estimated is not small. A common approach to this problem in human genetics is to use functional information to find causal variants or makers close to them. This paper shows that finding informative variants can be used to successfully increase genomic prediction accuracy in cattle.

Other attempts to find SNPs that improved prediction accuracy may have failed either because prediction was in the same breed as the training population and/or because there was little functional information on which to base selection of SNPs.

Karaman et al 2016. simulated one trait, body height, of humans using genotypes of a single ethnic group of 85 Caucasian individuals from the 2012 version of the 1000 human genome project and analysed over 30,000 SNPs. Our study analysed real phenotypic data of over 30 traits of over 100,000 cattle with multiple breeds across three countries, with 17 million sequence variants plus functional genomics data across many species. Given the significant differences in the design and scale of the two studies, they are expected to reach different conclusions.

In order to make these above points clear, we completely re-written the introduction of the manuscript and revised the Discussion to include more references from animal work, including Karaman et al., 2016. In the new introduction, we have acknowledged the fact that N_e plays an important part of the genomic prediction accuracy. However, this does not undermine our approach where we aim to find informative variants which can increase genomic prediction accuracy.

The study uses cow and bull data separately, as if they were different populations. The

rational was that the bulls are preselected. P-values for SNP based on cow and bull data were mostly very different. In fact, cows and bulls form one population, some of the cows' information was probably included in bulls' information as daughter information. Cows carry less information than bulls as bulls' predictions are based possibly on hundreds of daughters. Also, cows may be preselected as well. To maximize accuracy while minimizing biases, the authors could use methods that allow to use all genotyped animals together with ungenotyped animals, e.g., single-step Bayesian Regressions.

Author response: we acknowledge that the bull and cow are not different 'populations' and we have removed this wording throughout the manuscript.

There might be a misunderstanding of the reasons for analysing bulls and cows separately and we apologise if this was not clear enough in the previous manuscript. If SNPs are selected from a GWAS and then a prediction equation is estimated from the same data then a biased prediction can result. Consequently, we selected the SNPs in one sex and used them to derive a prediction equation in the other sex (selection was on SNPs not animals). A sentence 'If Bayes RC is applied to the same data as used to select the top 165k variants, this can result in bias because the non-significant variants are missing from the BayesRC analysis.' and some additional explanations have been added to the result section of '*Bayesian mixture modelling across sexes*'.

We agree with the reviewer that single-step Bayesian Regressions might increase genomic prediction accuracy but that is not the subject of this paper. We have included the reference of Fernando et al 2016 GSE in the 2nd paragraph of the introduction and 2nd last paragraph of Discussion. However, this does not mean that we should not use rich, existing and ever-growing multi-omics data (e.g., our FAETH score <https://www.pnas.org/content/116/39/19398>) to identify informative SNPs to increase genomic prediction accuracy, which we have demonstrated in the current study.

The authors use Cholesky decomposed traits. Such decomposition is impossible for 34 traits as the variance covariance matrix is certainly not positive definite, perhaps has a rank of 15, unless the authors "made it" positive definite. But then the results would make little sense for most of the transformed traits. There is an established technique for handling a large number of traits, possibly with a singular variance-covariance structure, called canonical transformation.

Author response: there appears to be some misunderstanding of our methods. The phenotype correlation matrices we used to do Cholesky decorrelation are not singular. If we had used estimated genetic correlation matrices they might well be singular, but as stated in the manuscript, we only used phenotypic correlation matrices. Cholesky trait-decorrelation is an established method as supported by our publications (e.g., Xiang et al 2017 <https://www.nature.com/articles/s41598-017-09788-9> and Xiang et al 2020 <https://www.nature.com/articles/s42003-020-0823-6>). This method allows us to use data from multiple traits with varying number missing values, thus increase the power of the study. We have expanded the explanation of these points in results (1st paragraph of Results) and Methods (1st paragraph is Methods). We could not use a canonical transformation because of the large amount of missing data.

Are accuracies in Figure 5 for bulls or cows? How are they derived? If for bulls, are they accuracies or reliabilities?

Author response: as stated in Results and Methods and the legend of Figure 5, only cow data were used for validation analysis. We have emphasized this point in the main legend of Figure 5.

The results are given only for 3 production traits. What about the other traits? It would be interesting to see results in other traits, e.g., fertility and productive life.

Author response: The animals used for validation did not have phenotypes for a complete range of traits so we estimated accuracies of prediction only for the production traits which were the most complete traits. To address the comments from the reviewer, we have assembled a new validation set of 6 phenotypes, including 3 production traits, 2 percentage traits and somatic cell count, of 14k Australian cows from three breeds. To meet different requests from different reviewers, we presented the results in two sections: 1) comparing the prediction accuracy of the final prioritisation of 80k variants (used to select markers on the XT50K SNP chip) with the markers from the XT50K panel, with a range of selections of top SNPs, as detailed in Supplementary Note S3 and briefly mentioned in the result section of '*Illustration of mapping and fine mapping of QTL*' in the main text; 2) compare the XT50K SNP chip with standard 50K and higher density panels such as HD SNP chip, as detailed in Supplementary Note S4 and briefly mentioned in the last paragraph of the result section in the main text. These additional data and analyses again supported our methods of selecting variants and the merit of the selected variants in the context of genomic prediction.

There is no legend in Figure 5, graph B. Are they for XT common with the US?

Author response: there appears to be another misunderstanding of the figure legend. Figure 5B had the legend in the initial submission. We are not able to evaluate the allele frequency of any SNPs in the US populations because we don't have the raw data and the GWAS summary statistics (Jiang et al 2019) don't have the allele frequency information. However, we have published the coordinates of the prioritised SNPs (Data S1) and interested readers can evaluate the frequency of these SNPs in their own data. We have also provided the allele frequency of the XT50K, standard50K and all sequence variants across different breeds in Australian animals in the Supplementary Figure S4 (briefly mentioned in the 1st paragraph of the result section of '*Final selection for inclusion on custom Infinium XT-50K beadchip*' in the main text).

The accuracies for NZ cattle based on US SNP data are much lower than those based on AU data. Why is it so? Is it because the US derived potential causative SNP are much poorer than those derived by AU? With standard 50k SNP, the accuracies for NZ Jerseys and their crosses in A & B should be the same and they are different.

Author response: When using the US GWAS marker effects to predict the Australian and New Zealand cows, the overall prediction accuracy for NZ traits is lower than the AU traits, even with the standard 50K panel. The same markers are used to predict the AU and NZ animals.. We can only attribute this to the potential fact that the genetic relationship between NZ and US animals is more distant than the relationship between Australian animals and US animals. The difference in the scale of phenotype between the NZ and US animals may also be larger than the difference between AU and US animals. These points have been added to the 8th paragraph of Discussion.

As stated in the manuscript (1st sentence in the section of '*Validation of the new XT-50K custom array for genomic prediction in global cattle data*'), the prediction accuracy results from Figure 5A, is based on BayesR training with multiple breeds. The prediction accuracy results from Figure 5B are based on GWAS effects with a single Holstein breed. We believe these differences are enough to explain the reviewer's question regarding the difference in prediction results of the Jersey breed. We have stressed this point in the discussion in the 8th paragraph of Discussion.

The paper uses many potential causative SNP. Based on experiences, most of these SNP have an effect too small to be well estimated. Can some accuracies be derived using only 10, 100 or 1000 variants with the biggest effect?

Author response: as described above, we have provided additional data to compare prioritised SNPs with other top selection which are detailed in Supplementary Note S3 and briefly mentioned in the result section of '*Illustration of mapping and fine mapping of QTL*' in the main text. The new results supported our method of selecting variants using local gEBV from BayesR to be the best. We also found that top variants with FAETH and BayesR PP ranking are the 2nd best, with the top variants selected by GWAS being the worst. Top 10k variant selections always had worse performances than top 80k variant selections in the context of genomic prediction accuracy. This tendency suggests that a very small number of variants will not have good prediction accuracies and therefore, we decide not to go any further to top 1000, or 100 variant selection.

Reviewer #3 (Remarks to the Author):

This paper outlines how methods to fine-map genome-wide informative sequence variants with with pleiotropic effects and functional significance were employed to select more

informative markers for a 50K SNP chip, and the resulting SNP chip improved the accuracy of genomic predictions in dairy cattle relative to the Standard-50K panel (BovineSNP50 beadchip, Illumina Inc). That is good news, but not entirely unexpected given the old Standard-50K panel was designed over a decade ago with basically no biological function data. But having said that the current analysis is a quite complete analysis using up-to-date data and biological information. It is limited to dairy evaluations and breeds though, or at least of the data presented. One major concern I have is that the authors just compared their XT-50K panel to the standard the Standard-50K panel (BovineSNP50 beadchip, Illumina Inc), and nothing else. The approach to generate the XT-50K is relatively complicated. It not clear how the XT-50K panel performance would compare to simpler methods? For example, just using the significant SNPs from BayesRC and some markers evenly distributed along the genome.

Author response: we thank the reviewer for the positive comments on our results. As noted by the reviewer, we have demonstrated a positive example of the integration of up-to-date functional, pleiotropy, genomic and extensive phenotype data to improve genomic prediction accuracy in dairy cattle data. The use of dairy cattle data is simply because there are many data in this sector. Many influential methods such as BayesR³ and BayesRC⁴ are firstly developed using dairy cattle data. Pleiotropic information can be identified using GWAS in any species^{5,6}. Results from FAETH ranking² can be also used to find informative variants in beef cattle. Our work is an important first step that proves that the combination of all these pieces of information can be used to increase in genomic selection. Novel knowledge and the analytical framework from this study can be influential to work in other farm animal species. Additional discussions of this point have been added to the 1st paragraph the Discussion. Also, to address requests from this and other reviewers, we have demonstrated the merit of our method of variant selection and the XT50K panel. A panel with a smaller number of SNPs has practical advantages and this panel has a much smaller number of SNPs but performed at least as well as denser panels such as GGP-F250 and HD. These points have been added to the first paragraph of the discussion to emphasise the significance of our work.

The reviewer asked if simpler SNP selection methods such as significant SNPs from BayesRC and some markers evenly distributed along the genome can work. In fact, what the reviewer suggested was not simple. First, completing BayesRC (after step 3 out of 4) already finished the majority of the hard work which started from 17 million SNPs. Also, we aim at providing a list SNPs that supported by biological data and/or variant trait associations which will be used in routine genotyping, not a panel with 'some markers evenly distributed along the genome'. However, to address the comments from the reviewer, we have conducted additional analysis where we trained 8 sets of variants selected differently using data of 6 traits of 28.1k Australian cows and predicted into additional 14.1k Australian cows with three breeds. Tested variant selections included top 10k and 80k variants with FAETH ranking, top 10k and 80k variants with GWAS ranking, top 10k and 80k with BayesRC posterior probability (PP) ranking, and our methods of BayesRC IgEBV and the final selection of XT50K SNPs. The results are shown in the Supplementary Note S3 and briefly mentioned in the result section of '*Illustration of mapping and fine mapping of QTL*' in the main text. The new results supported our method of selecting variants using local gEBV from BayesR to be the best. We also found that top variants with FAETH and BayesRC PP ranking are the 2nd best, with the top variants selected by GWAS being the worst.

Additionally, I was surprised to see no reference to the other bovine chips like 800K HD and GGP-F250? <https://link.springer.com/article/10.1186/s12711-019-0519-x> . This paper was not even referenced in the citations. At least a discussion comparing how this product

(https://genomics.neogen.com/pdf/slicks/ag229_ggp_f250_array_flyer.pdf) was developed and how the methods used to design the XT-50K panel were different/better would have been appropriate. Likewise how does the XT-50K compare to the improvement in accuracy from HD? The introduction had a number of statements about phenotype prediction and improved mapping with the XT-50K which is not what is contained in the paper. The authors need to define what they mean by precision breeding. This needs significant rewriting. The discussion and conclusion likewise need to reflect the data in the paper and not overstate the findings/outcome of this paper.

Author response: our study is trying to prioritise a set of variants that can be put on routinely used genotyping panels that usually have up to 50k markers. The HD panel that contains up to 800,000 SNPs panel is not a routinely used genotyping tool for tens of thousands of animals and we have stressed this point in the introduction. The panel of GGP-F250 contains up to 250,000 markers and the paper pointed out by the reviewer is published in late December 2019 when we already completed our manuscript. We have included this reference in the revised introduction.

Nevertheless, to address the request from the reviewer, we have conducted an additional analysis using data of 8 traits of 28.1k Australian cows and predicted into additional 14.1k Australian cows with three breeds, to compare the XT50K SNP chip with standard 50K, GGP-F250, and HD SNP chip. These results are detailed and discussed in Supplementary Note S4 and briefly mentioned in the last paragraph of the result section in the main text. We showed that the predictive power of the XT50K, with a much smaller number of markers, is at least as good as denser panels such as GGP-F250 and HD.

We apologize for the presentation of the writing. We have completely re-written the introduction and revised the manuscript as requested by the reviewer (see the following). We hope these efforts can address the concerns of the reviewer.

Specific comments

Line 1-2 Need to include bovine somewhere in the title – and also you are not predicting “traits” but rather **breeding values** for traits in cattle populations across countries

Author response: we agree with the reviewer that we are predicting breeding values. However, prediction of breeding value is called prediction of phenotype in the human genetics literature and we hoped this paper would be of interest to both hman and animal geneticists. We also try to shorten the title. Therefore, in the title we have retained the term ‘predict many traits’, but we have changed the term to predict breeding/genetic values in the text. The title now reads ‘Genome-wide fine-mapping identifies pleiotropic and functional variants that predict many traits across global cattle populations’

Line 10, 11 Is Cooperation really the correct name for this company? Abstract needs to be written in the past not present tense. We systematically fine-mapped etc for the whole abstract.

THERE ARE SPELLING ERRORS IN THE ABSTRACT!!!!!! NOT OK – run it through spell check - was this version really reviewed by all authors??

16 inforamtion spelled incorrectly

22 vertabrae spelled incorrectly

23 don’t use multiple twice in the same sentence – “numerous”? “a number of”

23 I do not see any data on improved mapping relative to the standard array– I would believe it could be true but was this done in the paper? I think perhaps two objectives are getting conflated

a. Demonstrate a method that can be used for genome-wide fine mapping of QTL, which was used b. Develop an improved 50K chip for more accurate prediction of breeding values

Author response: the co-authors confirmed with us that is their company name.

The tense of the abstract has been changed to the past.

We apologize for the typos and we have corrected them and checked throughout the manuscript to eliminate them.

We have removed redundant words 'multiple'

The term 'mapping' has been removed.

We thank the suggestion from the reviewer and replaced the last sentence of the abstract with 'We demonstrated a method for genome-wide fine-mapping of QTL and developed a biology-informed 50k chip for more accurate prediction of genetic values for many traits.'

25 substantiates is not the right word –“which can inform” “which may contribute to”

26 What is “precision breeding?” define this term This last sentence is problematic as I have no idea what it means.

Author response: these words/sentences have been removed in the revised manuscript.

30 replace “ones” with variants

34 “for precision medicine and agriculture where the prediction of phenotype is important”

When are we predicting phenotype in agriculture – usually predicting breeding value?

41 “precision breeding” is a term that is increasingly used for genome editing

<https://www.ncbi.nlm.nih.gov/pubmed/30835493> I have not seen it used before for customized genotyping aimed at increasing genomic selection accuracy. The authors write “customised genotyping for agriculture populations aiming at increasing genomic selection accuracy can be defined as a form of ‘precision breeding’.” Who else has defined the term this way? This just sounds to me like genomic selection with better genotypes, not “precision breeding”.

42 “Therefore, for accuracy in predicting phenotype” – is this what this paper is about? Or genomic selection accuracy??? These are VERY different goals

56 enriched for putative causal variants

57 The aim of this paper is to demonstrate a method that can be used both for genome-wide fine mapping of QTL and to select an informative panel of markers for prediction of phenotype. I don't believe that is what this paper does at all – it is a panel for genomic prediction NOT phenotype prediction. In fact on line 226 it says “We wish to identify a reduced panel of variants that could be used to predict the breeding value of all 34 traits.” NOT phenotype.

60 NOT phenotypes – breeding values

62 putative causal variants

66 Again mapping – where is data showing improved mapping of XT-50K markers versus standard 50K array?

Author response: as we have completely re-written the introduction, the above points have been already removed from the revised manuscript.

347 The improvement in the prediction accuracy for the US to Australia and New Zealand numbers were very high – as high as 90%. These numbers are high in part because the denominator of fat is very small (i.e. it is easier to double a very low number). I think a sentence indicating this would be appropriate to put the magnitude of the increase in context.

Author response: a sentence 'Genomic prediction accuracies of Jersey fat with the standard-50K markers were very low' has been added to the end of this section.

Also as I understood it this was looking at 27.7K of the Standard-50K panel to match the number of variants on the Australian XT-50K panel so I don't think it is correct to say "Comparing customized XT-50K markers with the Standard-50K markers in multiple validation populations" when in fact this is comparing 27.7K. – Perhaps should word it as "a subset of markers", or maybe I have misunderstood what was done.

Author response: the legend of Figure 5B was changed to 'Genomic prediction using a subset of markers from the two 50K panels that are also found in the US 36-trait GWAS using 27.1k Holstein US bulls for the same three traits and predicted into 28.2k Australian cows and 21.2k New Zealand cows.'

421 "While an enormous amount of effort is invested in genomics research, successful translation of genomic research outcomes into applications is rare." This is a VERY sweeping statement –especially given the impact of genomic selection, and disease variant identification. Please provide evidence to support this assertion – the standard 50K is a product of genomics research and is a rip-roaring success.

Author response: this sentence has been removed in the revised manuscript.

423 global dairy cattle breeding industry

Author response: dairy has been added to this sentence.

426 "Thus, many of the markers are likely to be causal or at least highly informative" I think this is overstating the data. Very few causal markers have been identified in the past 20 years. What evidence is there that these are causative markers? Informative perhaps, causative needs proof.

Delete or rephrase as "putative" causal variants

Author response: this sentence has been changed to 'Thus, many of them (the markers) can be *putatively* causal or at least highly informative.'

448 "precision breeding" – again what does this mean? Define

Author response: this sentence has been removed

452 "precise genomic selection" reword perhaps "more accurate genomic selection"

Author response: this sentence has been changed to 'we expect a chip panel to be highly enriched with causal variants which may be used in combination with single-step genomic prediction methods to allow more accurate genomic selection for many traits.'

459 prediction of multiple traits across dairy cattle populations and countries

462 "Genome wide fine-mapping plus causal-variant customised genotyping leads an innovative pathway to precision breeding." This sentence makes no sense because "precision breeding" is not defined – and really what this paper describes is an improved 50K SNP chip which increases the accuracy of genomic predictions; not "an innovative pathway to precision breeding" ". I really feel the language in the conclusions was a little overblown.

I think a better final sentence would read "The XT-50K panel improved the accuracy of genomic prediction accuracy in dairy cattle relative to the Standard-50K panel, especially in small breeds, and in across-breed and crossbred scenarios where prediction accuracies are usually lower."

Author response: the last sentence of the conclusion has been removed in the revised version.

465 the sentence "No live animals were used for this study" seems strange. Do you mean "The data analyzed in this study was collected previously [citations], and no further live animal experimentation was required for our analyses."

Author response: This sentence has been changed to ‘The data analyzed in this study was collected by DataGene Australia (<http://www.datagene.com.au/>) and no further live animal experimentation was required for our analyses.’

References:

1. Kemper, K.E. *et al.* Improved precision of QTL mapping using a nonlinear Bayesian method in a multi-breed population leads to greater accuracy of across-breed genomic predictions. *Genetics Selection Evolution* **47**, 29 (2015).
2. Xiang, R. *et al.* Quantifying the contribution of sequence variants with regulatory and evolutionary significance to 34 bovine complex traits. *Proceedings of the National Academy of Sciences* **116**, 19398-19408 (2019).
3. Erbe, M. *et al.* Improving accuracy of genomic predictions within and between dairy cattle breeds with imputed high-density single nucleotide polymorphism panels. *Journal of dairy science* **95**, 4114-4129 (2012).
4. MacLeod, I. *et al.* Exploiting biological priors and sequence variants enhances QTL discovery and genomic prediction of complex traits. *BMC genomics* **17**, 144 (2016).
5. Bolormaa, S. *et al.* A multi-trait, meta-analysis for detecting pleiotropic polymorphisms for stature, fatness and reproduction in beef cattle. *PLoS genetics* **10**, e1004198 (2014).
6. Xiang, R., van den Berg, I., MacLeod, I.M., Daetwyler, H.D. & Goddard, M.E. Effect direction meta-analysis of GWAS identifies extreme, prevalent and shared pleiotropy in a large mammal. *Commun Biol* **3**, 88 (2020).

REVIEWERS' COMMENTS

Reviewer #1 (Remarks to the Author):

The ms has improved in its legibility, although I still have the same concern in that there is lot of ad-hoc reasoning and on the difficulty in applying the same protocol to other cases. Nevertheless, if it works, that is fine.

How can it be that accuracy increases 30-60% in some breeds while 2% in other? This must be explained, perhaps verified.

Perhaps I missed it, could authors mention in ms how would they choose between a large effect SNP on a single trait and many small effects on several phenotypes?

Mention the nature of the 34 traits, milk yield, ... perhaps range of h^2 . In intro, mention also the three breeds used

the first sentence is abstract 'Causal mutations can affect gene expression, evolution, and many phenotypes' is kind of weird / truism. Something like 'finding causative mutations would help to improve prediction yet it is difficult' ...

line 83: better genotyping ? better prediction / performance ...

FAETH score is kind of inhouse score that is not standard like SIFT, may be describe briefly

Define local gEBV first time it appears in main text

Reviewer #3 (Remarks to the Author):

The manuscript is much improved and reads a lot better than the previous version. I have few remaining comments.

I would suggest adding the modifier "dairy" cattle populations to the title – there is no data in any traits other than dairy, and it is unclear how it might work on beef cattle.

Line 84 "better genotyping" should perhaps read "more accurate prediction of genetic merit"

Line 383 "We showed that the XT-50K, with up to 12-fold lower marker density, achieved at least a similar genomic prediction accuracy as the GGP-F250 and HD genotype panels." This seemed a strange phrasing. In the supplementary material it read "These results show that the predictive power of the XT-50K, with much smaller number of markers, is at least as good as denser panels such as GGP-F250 and HD. The standard 50K panel had the worst prediction accuracy across all scenarios." I would suggest rewording in the main manuscript to "The predictive power of the XT-50K, with much smaller number of markers, was similar and often better than denser panels such as GGP-F250 and HD. The standard 50K panel had the worst prediction accuracy across all scenarios."

Alison Van Eenennaam

We thank the reviewers for their time and effort in providing comments and suggestions on our manuscript. Please find our point-to-point response to reviewers' comments in blue text in the following.

Reviewer #1 (Remarks to the Author):

The ms has improved in its legibility, although I still have the same concern in that there is lot of ad-hoc reasoning and on the difficulty in applying the same protocol to other cases. Nevertheless, if it works, that is fine.

Author response: we thank the reviewer for the positive comments and thorough consideration of our work. We believe that we have done quite extensive extra analyses requested by the reviewer to support our results. We do not think our method is perfect. In fact, at the end of the 3rd paragraph of the Discussion we have already mentioned that 'However, it is likely we missed some informative variants due to the datasets used for this study.' On the other hand, as the first study of its kind, we do hope our study can lay a foundation for many future works to further explore the use of functional information in genomic prediction. We also provided extra discussions of our methodology as requested by the reviewer in the following responses.

How can it be that accuracy increases 30-60% in some breeds while 2% in other? This must be explained, perhaps verified.

Author response: we have added a new paragraph to discuss this issue (the 3rd last paragraph of the of discussion): 'When using the US Holstein GWAS to predict, the increase in the accuracy of XT-50K compared to Standard-50K panel in the Holstein cattle is smaller than such increase in accuracy in Jersey and Australian Red cattle (Figure 5B). The fact that there was no Jersey, Australian Red or mixed breeds in the US GWAS dataset limited its power in predicting traits of these two breeds in the Australian and New Zealand cows. This resulted in a lower accuracy for the Standard-50K to predict Jersey and Australian Red traits using the US Holstein GWAS, compared to the accuracy of predicting Holstein traits using the US Holstein GWAS. This would leave more room for the accuracy to improve when using US Holstein GWAS to predict Jersey and Australian Red traits with the XT-50K panel. Overall, the prediction accuracies using the GWAS summary statistics (Figure 5B) were much more variable than the accuracies using the raw data with BayesR model (Figure 5A). The use of GWAS summary statistics in genomic prediction (i.e., PRS in humans¹) of cattle complex traits requires more investigation in the future. Nevertheless, even though only around half of the traits overlapped between the Australian and the US data, putative pleiotropic variants affecting up to 36 US traits were strongly enriched in the Australian XT-50K panel but barely so for the Standard-50K panel (Figure 5C).'

Perhaps I missed it, could authors mention in ms how would they choose between a large effect SNP on a single trait and many small effects on several phenotypes?

Author response: our study selected variants that were functionally important and that they

show association with at least one trait in two sexes. Therefore, we don't particularly distinguish between variants with large effect on a single trait or variants with many small effects on multiple traits: based on multi-trait associations, these 2 types of variants will be both important and selected. We also select variants that had consistent ranking based on their functional importance and effects on one of 34 traits in bulls and cows within each chromosome segment. We added following text as the 2nd paragraph of the discussion:

The main feature of our variant prioritisation is the incorporation of functional information of variants not necessarily correlated with their size of effects on complex traits. Besides, variants were selected if they were associated with at least one trait (uncorrelated multi-trait GWAS^{2,3}, variant clustering with ρ_{ij} and variance of local gEBV summed over 34 traits). Also, we select variants that had consistent rankings based on their functionality and multi-trait associations between two sexes within each chromosome segment. This leads to the selected informative variants being evenly distributed across the genome, instead of being clustered in some parts of the genome.

Mention the nature of the 34 traits, milk yield, ... perhaps range of h2. In intro, mention also the three breeds used

Author response: we have modified the section towards the end of the last paragraph of the introduction as the following: '...this ideal and applies it to data on 44,000 dairy cattle from 3 breeds (**Holstein, Jersey and Australian Red**) with records on 34 traits **including milk production, fertility, management and body conformation (average h2 of 0.42±0.04 in bulls and 0.16±0.03 in cows)**. ...'

the first sentence is abstract 'Causal mutations can affect gene expression, evolution, and many phenotypes' is kind of weird / truism. Something like 'finding causative mutations would help to improve prediction yet it is difficult' ...

Author response: We thank the suggestion from the reviewer. We have modified the first sentence of the abstract which now reads 'The difficulty in finding causative mutations has hampered their use in genomic prediction.'

line 83: better genotyping ? better prediction / performance ...

Author response: We thank the suggestion from the reviewer. The selected variants are designed on a new SNP chip so we do expect a better genotyping to start with. Better genotyped initial data could then also lead to better genomic a prediction performance and we thus modified the sentence as: 'Finally, we derive a set of informative variants that we chose to be designed on a custom 50K array that will enable better genotyping **and more accurate prediction of genetic merit**, in many cattle.'

FAETH score is kind of inhouse score that is not standard like SIFT, may be describe briefly

Author response: we have modified the section towards the end of the last paragraph of the introduction as the following: '... .Firstly, we reduce the number of variants from 17M to 1.7M by carrying out a multi-trait GWAS using single variant regression incorporating the

Functional-And-Evolutionary Trait Heritability (FAETH) score, a publicly available ranking of cattle sequence variants based on their functionality and predicted heritability⁷. ...’

Define local gEBV first time it appears in main text

Author response: we have modified the sentence (1172 in the previous version) where local gEBV firstly appeared as such: ‘This prediction equation was also applied to the variants within each 50kb chromosomal segment to generate the genomic estimated breeding value for each 50kb segment, i.e., local gEBV, for 34 traits.’

=====

Reviewer #3 (Remarks to the Author):

The manuscript is much improved and reads a lot better than the previous version. I have few remaining comments.

Author response: we thank the positive comments from the reviewer.

I would suggest adding the modifier “dairy” cattle populations to the title – there is no data in any traits other than dairy, and it is unclear how it might work on beef cattle.

Author response: We agree with the reviewer that we did not analyse data from beef cattle. However, the title has reached the word-limit of the journal. To meet the reviewer in the middle, we have specified that our analysis used and validated in ‘dairy’ cattle in the revised abstract (accommodated some comments from the other reviewer and the editor). The two places in the abstract where we made correspondent changes are: ‘... Our analysis of 17 million sequence variants in 44,000+ Australian **dairy** cattle with 34 traits suggests on average one pleiotropic QTL existing in each 50kb chromosome-segment. ...’ and ‘This biology-informed custom array outperforms the standard array in predicting genetic value of multiple traits across populations in independent datasets of 90,000+ **dairy** cattle from the USA, Australia and New Zealand.’

Line 84 “better genotyping” should perhaps read “more accurate prediction of genetic merit”

Author response: We thank the suggestion from the reviewer. The selected variants will be (and are) implemented on a new SNP chip so we do hope a better genotyping using the new panel. Better genotyped initial data could then also lead to better genomic a prediction performance and we thus modified the sentence as: ‘Finally, we derive a set of informative variants that we chose to be designed on a custom 50K array that will enable better genotyping **and more accurate prediction of genetic merit**, in many cattle.’

Line 383 “We showed that the XT-50K, with up to 12-fold lower marker density, achieved at least a similar genomic prediction accuracy as the GGP-F250 and HD genotype panels.” This seemed a strange phrasing. In the supplementary material it read “These results show that the predictive power of the XT-50K, with much smaller number of markers, is at least as

good as denser panels such as GGP-F250 and HD. The standard 50K panel had the worst prediction accuracy across all scenarios.” I would suggest rewording in the main manuscript to “The predictive power of the XT-50K, with much smaller number of markers, was similar and often better than denser panels such as GGP-F250 and HD. The standard 50K panel had the worst prediction accuracy across all scenarios.”

Alison Van Eenennaam

Author response: we thank the reviewer for the suggestion, and we have changed the sentence to the version suggested by the reviewer (at end of the last paragraph of Results).

1. Wray, N.R., Kemper, K.E., Hayes, B.J., Goddard, M.E. & Visscher, P.M. Complex Trait Prediction from Genome Data: Contrasting EBV in Livestock to PRS in Humans: Genomic Prediction. *Genetics* **211**, 1131-1141 (2019).
2. Xiang, R., MacLeod, I.M., Bolormaa, S. & Goddard, M.E. Genome-wide comparative analyses of correlated and uncorrelated phenotypes identify major pleiotropic variants in dairy cattle. *Scientific Reports* **7**, 9248 (2017).
3. Xiang, R., van den Berg, I., MacLeod, I.M., Daetwyler, H.D. & Goddard, M.E. Effect direction meta-analysis of GWAS identifies extreme, prevalent and shared pleiotropy in a large mammal. *Commun Biol* **3**, 88 (2020).